# Cells of the adult human heart

Monika Litviňuková[1,2,21], Carlos Talavera-López[1,3,21], Henrike Maatz[2,21], Daniel Reichart[4,5,21], Catherine L. Worth[2], Eric L. Lindberg[2], Masatoshi Kanda[2,6], Krzysztof Polanski[1], Matthias Heinig[7,8], Michael Lee[9], Emily R. Nadelmann[4], Kenny Roberts[1], Liz Tuck[1], Eirini S. Fasouli[1], Daniel M. DeLaughter[4], Barbara McDonough[4,11,16], Hiroko Wakimoto[4], Joshua M. Gorham[4], Sara Samari[9], Krishnaa T. Mahbubani[12], Kourosh Saeb-Parsy[12], Giannino Patone[2], Joseph J. Boyle[9], Hongbo Zhang[1,13], Hao Zhang[14,15], Anissa Viveiros[14,15], Gavin Y. Oudit[14,15], Omer Ali Bayraktar[1], J. G. Seidman[4,22✉], Christine E. Seidman[4,11,16,22✉], Michela Noseda[9,17,22✉], Norbert Hubner[2,10,18,19,22✉] & Sarah A. Teichmann[1,20,22✉]

Cardiovascular disease is the leading cause of death worldwide. Advanced insights into disease mechanisms and therapeutic strategies require a deeper understanding of the molecular processes involved in the healthy heart. Knowledge of the full repertoire of cardiac cells and their gene expression profiles is a fundamental first step in this endeavour. Here, using state-of-the-art analyses of large-scale single-cell and single-nucleus transcriptomes, we characterize six anatomical adult heart regions. Our results highlight the cellular heterogeneity of cardiomyocytes, pericytes and fibroblasts, and reveal distinct atrial and ventricular subsets of cells with diverse developmental origins and specialized properties. We define the complexity of the cardiac vasculature and its changes along the arterio-venous axis. In the immune compartment, we identify cardiac-resident macrophages with inflammatory and protective transcriptional signatures. Furthermore, analyses of cell-to-cell interactions highlight different networks of macrophages, fibroblasts and cardiomyocytes between atria and ventricles that are distinct from those of skeletal muscle. Our human cardiac cell atlas improves our understanding of the human heart and provides a valuable reference for future studies.

The heart is a complex organ, composed of four morphologically and functionally distinct chambers (Fig. 1a). Deoxygenated blood from the low-pressure right atrium and ventricle is propelled into the lungs. Oxygenated blood enters the left atrium and ventricle, which propels blood across the body at systemic pressure. Chambers are separated by the interatrial and interventricular septa, and unidirectional flow is established by the atrio-ventricular and ventricular-arterial valves. An intrinsic electrophysiological system rapidly propagates electrical impulses from the sinoatrial node to the atrioventricular node, and along Purkinje fibres to the apex where contraction begins. Cardiac anatomical and functional complexity requires exquisite orchestration of heterogeneous cell populations to enable continuous contraction and relaxation across different pressures, strains and biophysical stimuli in each chamber.

The heart is derived from multipotent progenitor cells that comprise two heart fields. Cells of the first heart field primarily populate the left ventricle; second heart field cells populate the right ventricle, and both fields contribute to the atria. Haemodynamics changes in the postnatal period and the distinct gene regulatory networks that operate in each heart field presumably prime gene expression patterns of adult heart cells[1].

Single-cell and single-nucleus RNA sequencing (scRNA-seq and snRNA-seq, respectively) and multiplex single-molecule fluorescence in situ hybridization (smFISH) enable the identification of anatomical specificities, molecular signatures, intercellular networks and spatial relationships by illuminating the coordinated communication of cardiac cells within their microenvironments[2].

We present comprehensive transcriptomic data on six distinct cardiac regions, providing, to our knowledge, the largest reference framework so far[3,4]. We incorporate snRNA-seq to ensure high-throughput capture of large cardiomyocytes (length and width approximately

[1]Cellular Genetics Programme, Wellcome Sanger Institute, Wellcome Genome Campus, Hinxton, UK. [2]Cardiovascular and Metabolic Sciences, Max Delbrück Center for Molecular Medicine in the Helmholtz Association (MDC), Berlin, Germany. [3]EMBL - EBI, Wellcome Genome Campus, Hinxton, UK. [4]Department of Genetics, Harvard Medical School, Boston, MA, USA. [5]Department of Cardiology, University Heart & Vascular Center, University of Hamburg, Hamburg, Germany. [6]Department of Rheumatology and Clinical Immunology, Sapporo Medical University, Sapporo, Japan. [7]Institute of Computational Biology (ICB), HMGU, Neuherberg, Germany. [8]Department of Informatics, Technische Universitaet Muenchen (TUM), Munich, Germany. [9]National Heart and Lung Institute, Imperial College London, London, UK. [10]DZHK (German Centre for Cardiovascular Research), Partner Site Berlin, Berlin, Germany. [11]Cardiovascular Division, Brigham and Women's Hospital, Boston, MA, USA. [12]Department of Surgery, University of Cambridge, NIHR Cambridge Biomedical Centre, Cambridge Biorepository for Translational Medicine, Cambridge, UK. [13]Department of Histology and Embryology of Zhongshan School of Medicine, Sun-Yat Sen University, Guangzhou, China. [14]Division of Cardiology, Department of Medicine, Faculty of Medicine and Dentistry, University of Alberta, Edmonton, Alberta, Canada. [15]Mazankowski Alberta Heart Institute, Faculty of Medicine and Dentistry, University of Alberta, Edmonton, Alberta, Canada. [16]Howard Hughes Medical Institute, Chevy Chase, MD, USA. [17]British Heart Foundation Centre of Regenerative Medicine, British Heart Foundation Centre of Research Excellence, Imperial College London, London, UK. [18]Charité-Universitätsmedizin, Berlin, Germany. [19]Berlin Institute of Health (BIH), Berlin, Germany. [20]Deptartment of Physics, Cavendish Laboratory, University of Cambridge, Cambridge, UK. [21]These authors contributed equally: Monika Litviňuková, Carlos Talavera-López, Henrike Maatz, Daniel Reichart. [22]These authors jointly supervised this work: J. G. Seidman, Christine E. Seidman, Michela Noseda, Norbert Hubner, Sarah A. Teichmann. ✉e-mail: seidman@genetics.med.harvard.edu; cseidman@genetics.med.harvard.edu; m.noseda@imperial.ac.uk; nhuebner@mdc-berlin.de; st9@sanger.ac.uk

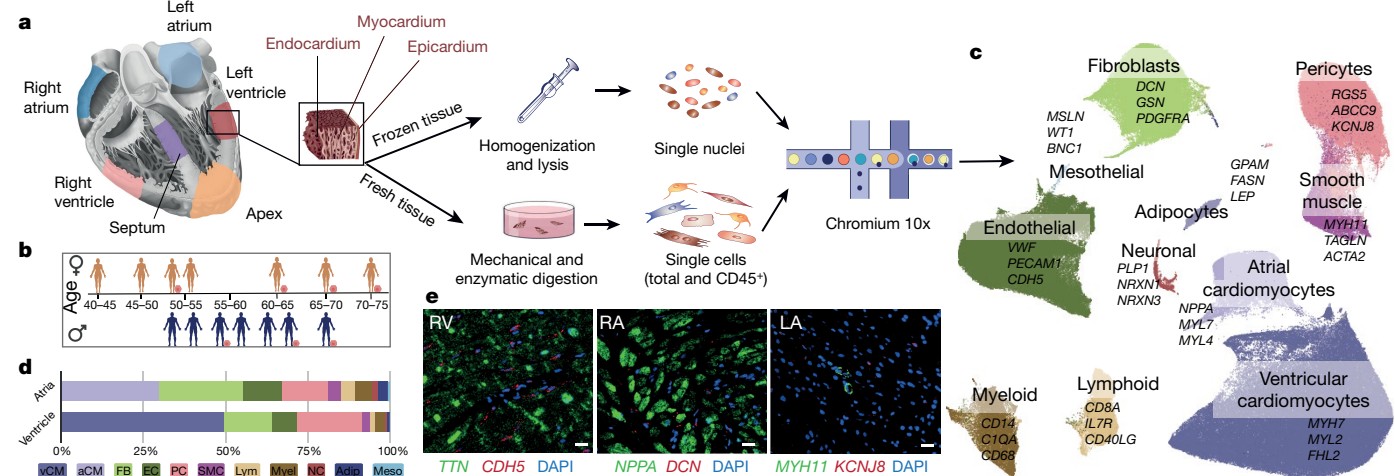

**Fig. 1 | Cell composition of the adult human heart. a**, Transmural samples were obtained from left and right atrium, left and right ventricles, apex and interventricular septum from 14 individuals. Single nuclei (n = 14) and single cells (n = 7) were processed using Chromium 10x 3'DEG chemistry. **b**, Infographic shows donors (women, top; men, bottom), age, and contribution to cells and nuclei datasets (orange circle). Data are available in Supplementary Table 1. **c**, Uniform manifold approximation and projection (UMAP) embedding of 487,106 cells and nuclei delineate 11 cardiac cell types and marker genes. **d**, Distribution of cell populations, identified from nuclei within atria (left and

right) and ventricles (left, right, apex and interventricular septum) after subclustering analysis. Colour code as in **c**. Data are available in Supplementary Table 2. Adip, adipocytes; Lym, lymphoid; Meso, mesothelial cells; Myel, myeloid; NC, neuronal cells; PC, pericytes. **e**, Multiplexed smFISH of cell type-specific transcripts in right ventricles (RV; left): *TTN* (green, cardiomyocytes) and *CDH5* (red, EC) right atrium (RA; middle): *NPPA* (green, aCM) and *DCN* (red, FB) and left atrium (LA; right): *MYH11* (green, SMCs) and *KCNJ8* (red, pericytes). Nuclei are counterstained with DAPI (dark blue). Scale bars, 20 μm. For details on statistics and reproducibility, see Methods.

100 and 25 μm) and scRNA-seq to upsample and enrich endothelial and immune cell populations. Using multiplex smFISH imaging, we describe the spatial distribution of selected cell populations and cell–cell co-localizations. We compare cardiac cell and nuclear transcriptomes with those of skeletal muscle and kidney, highlighting cardiac-specific cell signatures. Our study defines the cellular and molecular signatures of the adult healthy heart, and enables functional plasticity in response to varying physiological conditions and diseases.

## Cellular landscape of the adult human heart

We isolate single cells, nuclei and CD45+ enriched cells from the left and right ventricular free walls, left and right atrium, the left ventricular apex, and interventricular septum, from 14 adults (Fig. 1a, b, Supplementary Table 1). After processing with 10X Genomics and a generative deep variational autoencoder, the resulting dataset comprises 45,870 cells, 78,023 CD45+ enriched cells and 363,213 nuclei for 11 major cell types: atrial cardiomyocytes, ventricular cardiomyocytes, fibroblasts (FBs), endothelial cells (ECs), pericytes, smooth muscle cells (SMCs), immune cells (myeloid and lymphoid), adipocytes, mesothelial cells and neuronal cells (Fig. 1c, e, Extended Data Figs. 1, 2).

The distributions of these main cell types, estimated from nuclei data, differ between atrial and ventricular tissues. Atrial tissues contain 30.1% cardiomyocytes, 24.3% FBs, 17.1% mural cells (pericytes and SMCs), 12.2% ECs and 10.4% immune cells (myeloid and lymphoid). By contrast, ventricular regions (apex, interventricular septum, left and ventricle) contain 49.2% ventricular cardiomyocytes, 21.2% mural cells, 15.5% FBs, 7.8% ECs and 5.3% immune cells (Fig. 1d, Supplementary Table 2).

The ventricular proportions of ventricular cardiomyocytes and FBs are negatively correlated, whereas pericytes and SMC proportions are positively correlated, indicating a functional organization (Supplementary Table 3). Cell distributions are generally similar in male and female hearts. However, the mean percentages of ventricular cardiomyocytes from left and right ventricles are higher in female hearts (56 ± 9%; mean ± s.d.) and associated with a stronger negative correlation between ventricular cardiomyocytes and FBs (r = −0.8; slope = −0.9) compared to male hearts (47 ± 11%; P = 0.03;

ventricular cardiomyocytes–FBs, r = −0.4; slope = −0.3). Differences in the proportions of cardiomyocytes is unexpected given the average smaller female heart mass, and if confirmed might explain higher cardiac stroke volumes in women[5] and lower rates of cardiovascular disease.

## Cardiomyocyte heterogeneity

Cardiomyocytes show high-level expression of genes that encode contractile force-generating sarcomere proteins (*TTN*, *MYBPC3* and *TNNT2*) and calcium-mediated processes (*RYR2*, *PLN* and *SLC8A1*). Consistent with bulk tissue RNA-sequencing (RNA-seq) data[6], we observe markedly distinct transcriptional signatures in ventricular and atrial cardiomyocytes, reflecting developmental origins and differences in electrophysiological, contractile and secretory processes (Extended Data Fig. 3, Supplementary Table 4).

Ventricular cardiomyocytes are enriched in genes encoding sarcomere proteins (*MYH7* and *MYL2*), transcription factors (*IRX3*, *IRX5*, *IRX6*, *MASP1* and *HEY2*), and *PRDM16*, mutated in left ventricular non-compaction cardiomyopathy[7]. Other abundant transcripts enable tissue integrity despite high ventricular strain: *PCDH7* encodes a calcium-dependent strong adhesive molecule[8]; *SMYD2* encodes a lysine methyltransferase that promotes sarcomere formation and stabilization[9]. Atrial cardiomyocytes abundantly express prototypic genes and also *ALDH1A2*, an enzyme required for retinoic acid synthesis, *ROR2*, which participates in Wnt signalling during lineage differentiation[10], and *SYNPR*, which functions in the mechanosensing of TRP channels by atrial volume receptors[11].

We identify five ventricular cardiomyocyte (vCM1–vCM5) populations: vCM1 comprise 63.9% of left ventricular cardiomyocytes but only 36.7% of right ventricular cardiomyocytes (Fig. 2a, b, Extended Data Fig. 3a, c–e, Supplementary Tables 5, 6). vCM2 is more enriched in right ventricles (39.9%) than left ventricles (9.1%). However, differences between vCM1 and vCM2 are small, indicating shared gene programs between left (enriched in vCM1) and right (enriched in vCM2) ventricles. vCM2 shows higher expression of *PRELID2*, a developmental molecule with unknown cardiac function[12] (verified by single-molecule

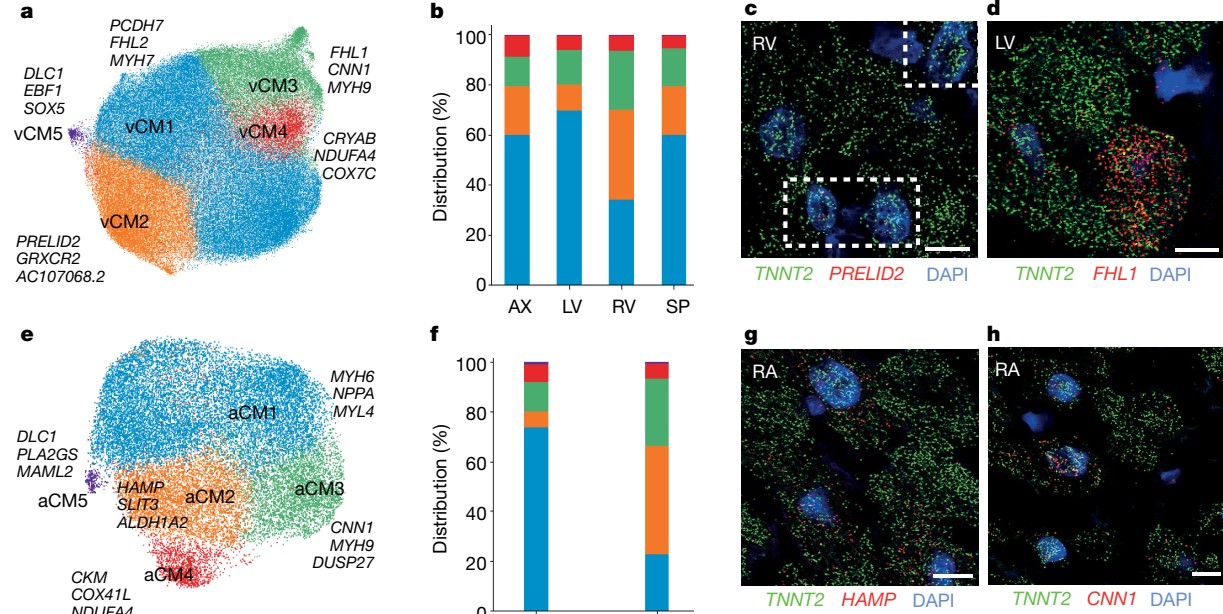

**Fig. 2 | Cardiomyocytes. a**, UMAP embedding of five ventricular cardiomyocyte (vCM) populations. **b**, Regional distributions of ventricular cardiomyocyte populations. Data are available in Supplementary Table 5. AX, apex; LV, left ventricle; SP, interventricular septum; RV, right ventricle. **c**, **d**, Multiplexed smFISH of *PRELID2* (red) enriched in vCM2 (**c**) and of *FHL1* (red) enriched in vCM3 (**d**). **e**, UMAP embedding of five atrial cardiomyocytes

(aCM) populations. **f**, Regional distributions of atrial cardiomyocyte populations. LA, left atrium; RA, right atrium. **g**, **h**, Multiplexed smFISH of *HAMP* (red) enriched in aCM2 (**g**) and of *CNN1* (red) enriched in aCM3 (**h**). In **c**, **d**, **g** and **h**, nuclei are counterstained with DAPI (dark blue). Scale bars, 10 μm. For details on statistics and reproducibility, see Methods.

fluorescence in situ hybridization (smFISH) (Fig. 2c, Extended Data Fig. 3c). Among vCMs, vCM2 has the highest expression of the myosin gene *MYH6* and *CDH13*, a cell surface T-cadherin receptor for cardioprotective adiponectin and low density lipoproteins, both associated with several cardiometabolic traits[13].

vCM3 and vCM4 are present across all ventricular regions. The vCM3 transcriptional profile resembles a prominent right atrium population (aCM3, discussed below) with retinoic-acid-responsive SMC gene enrichment (*MYH9*, *NEXN* and *CNN1*)[14,15]. vCM3 also express stress-response genes including *ANKRD1*[16], *FHL1*[17] (verified by smFISH) (Fig. 2d, Extended Data Fig. 3c), *DUSP27*[18], and *XIRP1* and *XIRP2*, interacting with intercalated disc ion channel proteins implicated in cardiomyopathy and arrhythmias[19]. The small population vCM4 is enriched for nuclear-encoded mitochondrial genes (*NDUFB11*, *NDUFA4*, *COX7C* and *COX5B*) and Gene Ontology terms indicative of a high energetic state (Extended Data Fig. 3f). vCM4 also demonstrates high levels of *CRYAB*, which encodes a cytoprotective and antioxidant heat shock protein[20], of sarcomere protein genes and *PLN*, indicating that these ventricular cardiomyocytes are outfitted to perform higher workload than other ventricular cardiomyocytes.

vCM5 (approximately 1%) comprises cells with high levels of *DLC1* and *EBF2*[21], regulating brown adipocyte differentiation and perhaps cardiac pacemaker activity, and transcripts identified in neuronal lineages (*SOX5*, *EBF1* and *KCNAB1*). As *EBF1*-depleted mice have a profoundly hypoplastic ventricular conduction system[22], vCM5 may participate in electrophysiology.

We identify five atrial cardiomyocyte populations (aCM1–aCM5) (Fig. 2e, f, Extended Data Fig. 3b–d, Supplementary Tables 6, 7). *HAMP*, a master regulator of iron homeostasis, is considerably enriched in more than 50% of right atrium cardiomyocytes versus 3% left atrium cardiomyocytes (verified by smFISH)[23] (Fig. 2g, Extended Data Fig. 3c), indicating energetic differences[6]. *HAMP* has unknown roles in cardiac biology, but *Hamp*-null mice have deficits in the electron transport chain and lethal cardiomyopathy[24].

aCM1 shows robust expression of prototypic atrial transcripts, indicative of a basal atrial cardiomyocyte gene program, and lower levels of molecules with neuronal functions (*ADGRL2*, *NFXL1* and *ROBO2*). aCM2 predominantly expresses *HAMP* within the right atrium and is enriched for *SLIT3*, the developmental ligand for cardiac ROBO receptors[25], *ALDH1A2*[26] and *BRINP3*, involved in retinoic acid signalling, and *GRXCR2*, supporting cilia involved in mechanosensing[27].

aCM3 and vCM3 share similar transcriptional profiles including enrichment of the SMC gene *CNN1* (verified by smFISH) (Fig. 2h, Extended Data Fig. 3c). The molecular signatures of aCM2, aCM3 and vCM3 indicate derivation from the second heart field[28]. aCM4 transcripts denote high metabolic activity, similar to vCM4, and have the highest expression of *NPPA*. aCM5 expresses similar transcripts to vCM5.

## Vascular, stromal and mesothelial cells

The vascular compartment includes 17 distinct populations of EC, SMC, pericyte, mesothelial cells with anatomical and arterio-venous specificities (Fig. 3a, b, Supplementary Tables 8, 9). Endothelial cells, identified by pan-EC markers *PECAM1*, *CDH5* and *VWF*, comprise 10 populations (Extended Data Fig. 4a–c, g). Three capillary ECs (EC1-3_cap), which express *RGCC* and *CA4*[29], represent 57.4% of all ECs. Capillary-like EC4_immune ECs express transcripts related to antigen presentation and immune regulation (*CX3CL1*, *CCL2*, *IL6* and *ICAM1*)[30]. Arterial EC5_art ECs are enriched for *SEMA3G*, *EFNB2* and *DLL4*, whereas EC6_ven ECs express venous markers *NR2F2*[31] and *ACKR1*[32], which we confirmed by smFISH (Fig. 3c). Mainly atrial EC7_atria ECs express the angiogenesis regulator *SMOC1*[33] and *NPR3*, detected in mouse endocardium[34], suggestive of endocardial cells. Lymphatic EC8_ln ECs, enriched for *PROX1*, *TBX1* and *PDPN*, represent approximately 1% of the captured ECs[29].

Pericytes express *ABCC9* and *KCNJ8* and segregate into four clusters, with PC1_vent cells enriched in ventricles and PC2_atria cells

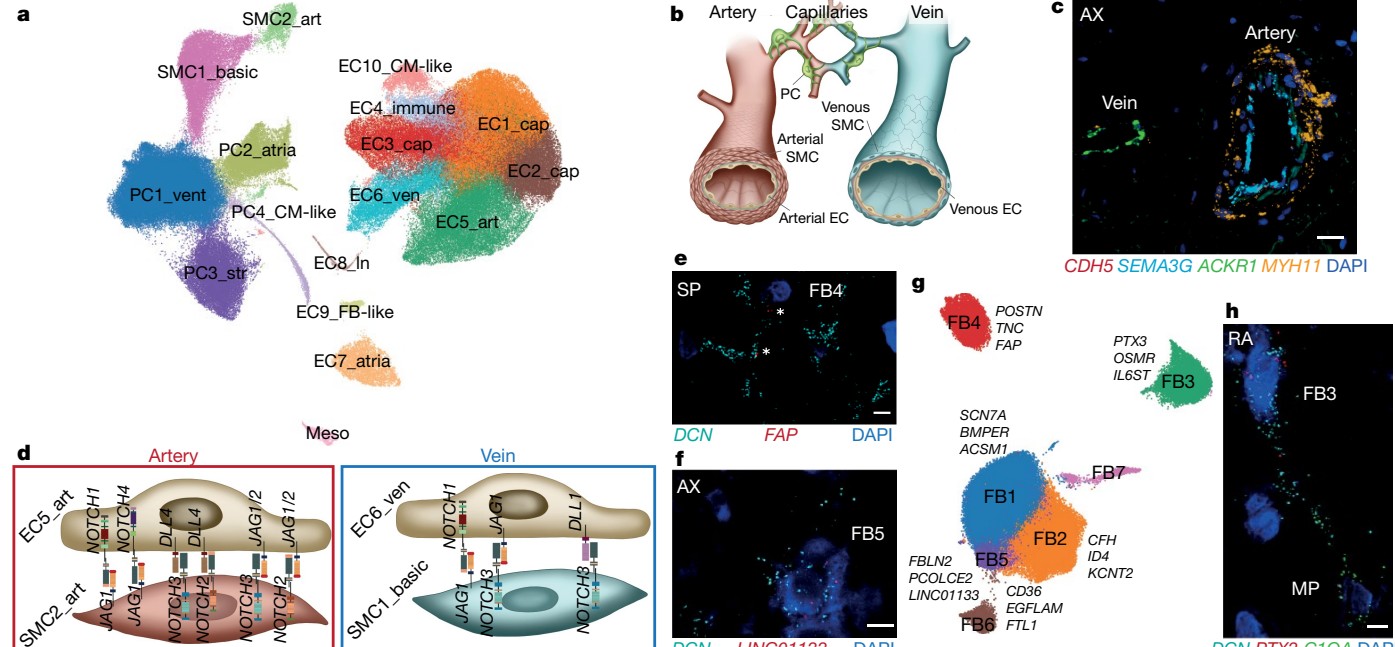

**Fig. 3 | Vascular, stromal and mesothelial cells. a**, UMAP embedding of 17 vascular and mesothelial populations. EC1/2/3_cap, capillary ECs; EC4_immune, immune-related ECs; EC5_art, arterial ECs; EC6_ven, venous ECs; EC7_atrial, atria-enriched ECs; EC8_ln, lymphatic ECs; EC9_FB-like, ECs with FB features; EC10_CM-like, ECs with cardiomyocyte features; PC1_vent, ventricle-enriched pericytes; PC2_atrial, atria-enriched pericytes; PC3_str, stromal pericytes; PC4_CM-like, pericytes with cardiomyocyte features; SMC1_basic, basic SMCs; SMC2_art, arterial SMCs. **b**, Schematic of the vascular cells and their placement in the vasculature. **c**, Multiplexed smFISH of *MYH11* (yellow) in SMC (thick in artery and very thin in small vein), *CDH5* (red) in the endothelium, and *SEMA3G* (cyan) and *ACKR1* (green) in EC5_art and EC6_ven, respectively in apex. Nuclei are counterstained with DAPI (dark blue). Scale bar, 20 μm. **d**, Predicted cell–cell interactions in arteries and veins. Data are available in Supplementary Table 10. **e**, **f**, Multiplexed smFISH of pan-FB *DCN* (cyan) and *FAP* (red) in FB4 in interventricular septum (SP) (**e**) and *DCN* (cyan) and *LINC001133* (red) in FB5 in the apex (AX) (**f**). Nuclei are counterstained with DAPI (dark blue). Scale bars, 5 μm. **g**, UMAP embedding showing six FB populations and their respective marker genes. **h**, Multiplexed smFISH of *C1QA*+ macrophages (MP) and *PTX3*+ FB3, suggesting cross-talk between both cell types. Scale bar, 5 μm. For details on statistics and reproducibility, see Methods.

in atria. PC1_vent cells express adhesion molecules (*NCAM2* and *CD38*), and *CSPG4*, which is involved in microvascular morphogenesis and EC cross-talk[35] (Extended Data Fig. 4d–f). PC3_str co-express pericyte markers and very low levels of pan-EC transcripts. RNA velocity analyses suggest a directionality that indicates PC3_str cells as a transitional state between pericytes and ECs (Extended Data Fig. 4h, i). These observations may relate to bidirectional pericyte or endothelial cell (trans)differentiation, which remains controversial[36].

Vascular SMCs that express *MYH11* split into two populations. SMC1_basic cells express transcripts that indicate immaturity, including the stem-cell marker *LGR6*[37] and proliferation-associated *RGS5*[38]. SMC2_art cells express considerably higher levels of *CNN1*, *ACTA2* and *TAGLN*, indicating arterial origin, whereas SMC1_basic cells may be venous-derived[39] (Extended Data Fig. 4d–f).

Cell–cell interaction analyses indicate connections between ECs and mural cells in different vascular segments (Fig. 3b–d, Extended Data Fig. 4j, k, Supplementary Table 10), including Notch receptor–ligand interactions[40] (NOTCH1 or NOTCH4 with JAG1, and NOTCH2 or NOTCH3 with JAG1, JAG2 or DLL4) between EC5_art and SMC2_art cells. Exploratory spatial transcriptomics (Extended Data Fig. 5) shows co-occurrence of EC5_art and SMC2_art markers and *JAG1* and *NOTCH2*, thereby supporting this interaction between ECs and SMCs. A venous-specific DLL1–NOTCH3 interaction is predicted for EC6_ven and SMC1_basic cells. Notably, many of the venous and arterial EC predicted interactions are shared with capillary ECs, which suggests gradual changes along the arterio-venous axis[39].

We define a distinct small population as mesothelial cells that enrich for *MSLN*, *WT1* and *BNC1*[41] but lack EC, FB or mural genes. smFISH confirms this annotation with localization of BNC1+/CDH5− cells to the epicardium (Extended Data Fig. 4l–n).

## Cardiac fibroblasts

Cells of the FB compartment show enriched expression of *DCN*, *GSN* and *PDGFRA* within seven populations (Fig. 3g) with regional enrichment in ventricles (FB1) and atria (FB2). This is consistent with distinctive functional properties, including stronger profibrotic responses, by atrial FBs[42]. FB1 and FB2 cells express canonical genes and define a basal, chamber-specific FB expression program (Extended Data Fig. 6a, Supplementary Table 11).

FB4 and FB5 cells are less abundant in the right atrium than other regions, whereas FB3 are less abundant in the left ventricle (Extended Data Fig. 6c). FB4 cells express genes responsive to TGFβ signalling (for example, *POSTN* and *TNC*) (Fig. 3e). FB5 cells have higher expression of genes involved in the production, remodelling and degradation of extracellular matrix (ECM). By contrast, FB3 cells have lower expression of ECM-related genes but higher expression of cytokine receptors such as *OSMR* and *ILST6*[43] (Fig. 3f, Extended Data Fig. 6b, Supplementary Table 12). These distinctive fibroblast gene programs probably govern stress-responsive cardiac remodelling and contribute to homeostasis.

Separate clustering of atrial and ventricular FBs recapitulated the populations described above, including an OSM-signalling population in each chamber (aFB4 and vFB3). In addition, we identify distinct chamber-specific ECM-producing FBs that differ in the expression of collagen isoforms and other ECM-related (aFB2 versus vFB2) (Extended Data Fig. 6g–m, Supplementary Table 13) or connective tissue-related genes (aFB1 versus vFB4).

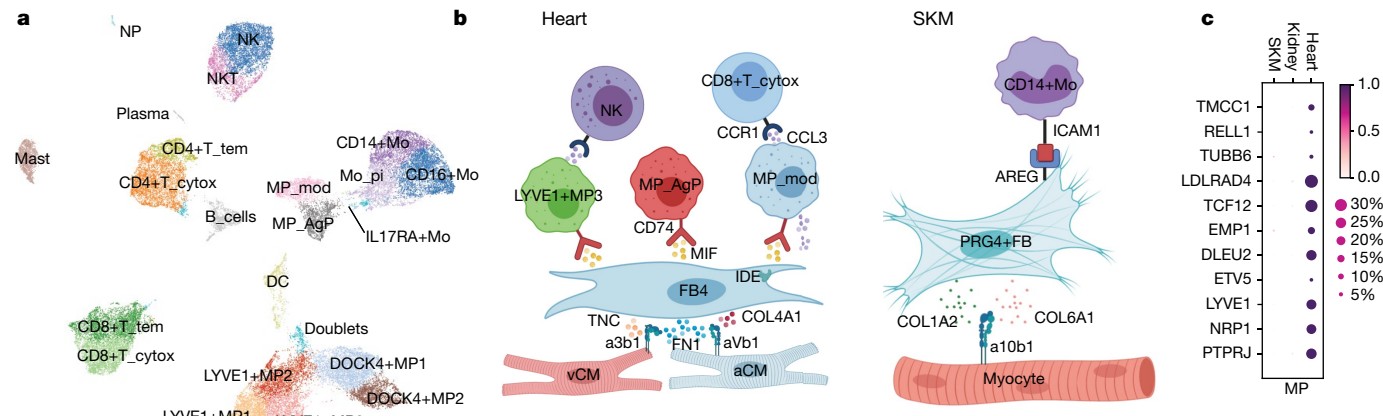

**Fig. 4 | Cardiac immune populations and cell–cell interactions. a**, Manifold of 40,868 myeloid and lymphoid cardiac cells. NP, neutrophils; NK, natural killer; NKT, natural killer T cells; CD4+T_tem, effector-memory CD4+ T cells; CD4+T_cytox, CD4+ cytotoxic T cells; CD8+T_tem, CD8+ effector-memory T cells; CD8+T_cytox, CD8+ cytotoxic T cells; DC, dendritic cells; CD14+Mo, CD14+ monocytes; CD16+Mo, CD16+ monocytes; Mo_pi, pro-inflammatory monocytes; IL17RA+Mo, IL17RA+ monocytes; MP_AgP, HLA class II antigen-presenting macrophages; MP_mod, monocyte-derived macrophages; LYVE1+MP1–3, M2-like, LYVE1+ macrophages sets 1–3; DOCK4+MΦ1–2, *DOCK4+* macrophage sets 1–2; B_cells, B cells; plasma, plasma B cells. **b**, BioRender infographic summarizes predicted cell–cell interaction circuits between atrial and ventricular cardiomyocytes, FB4 and immune cells involved in tissue repair in the heart and SKM. Data are available in Supplementary Table 17. **c**, Gene expression signature for cardiac-specific *LYVE1+* macrophages compared against predicted matched populations in skeletal muscle and kidney.

## Immune cells and cardiac homeostasis

Analysis of cardiac immune cells reveals 21 cell states (Fig. 4a, Extended Data Fig. 7). Myeloid cells comprise 13 populations, including several subtypes of macrophages, monocytes and dendritic cells, whereas the lymphoid compartment comprises 8 populations (Supplementary Tables 14, 15).

Macrophages include three *LYVE1+* macrophage populations: *LYVE1+* MP1–2 enrich for clathrin and cathepsin genes, and *LYVE1+* MP3 for *HLA-DOA*, *HLA-DQA1/2* and *HLA-DQB1*. *LYVE1+* macrophages appear related to recently described tissue-resident macrophages associated with cardiovascular remodelling[44], although negative for *TIMD4*[45] (Extended Data Fig. 8a, b). Monocyte-derived macrophages express *LYVE1* and *FOLR2*, monocyte-like markers *CEBPB* and *S100A8*, and chemoattractant cytokine genes *CCL13* and *CCL18*. Antigen-presenting macrophages are *FOLR2−*, *LYVE1−* and *MERTK−*, and enrich for *HLA-DRA*, *HLA-DMA*, *HLA-DMB*, *HLA-DPA1* and *TREM2* (described in lipid-associated macrophages)[46]. Although monocytes are abundant in our data and others[47], these are likely to be circulating, as supported by computational integration of our data with published peripheral blood mononuclear cell (PBMC) single-cell RNA-seq data (Supplementary Table 16). Two populations of *DOCK4+* macrophages differentiated by higher expression of *IL4R*, *STAT3* and *ITGAM* in *DOCK4+* MP1 versus *DOCK4+* MP2, do not express *C1QA* or *FOLR2* (Extended Data Fig. 8c).

Predicted cell–cell interactions identify receptor–ligand circuits among immune cells, cardiomyocytes, and FBs. *LYVE1+*, monocyte-derived and antigen-producing macrophages are predicted to interact with FB4 via CD74–MIF (Fig. 4b, Extended Data Fig. 8d, e, Supplementary Table 17). Inhibition of this interaction leads to fibrosis[48] and tissue damage[49]. FB4 also enrich for *FN1*, *COL4A1* and *TNC*, facilitating cellular proliferation in the fetal heart[50] and predicted to interact with different integrins in atrial and ventricular cardiomyocytes. In skeletal muscle (SKM), predicted cell–cell interactions between *PRG4+* FBs (analogous to FB4) and cardiomyocytes involve COL1A2, COL6A2 and α10β1 integrins, whereas SKM FBs and monocytes appear to interact via the ICAM1–AREG and CXCR4–CXCL12 chemokine pairs (Fig. 4b, Supplementary Table 18), indicating tissue-specific homeostatic transcriptional circuits.

Using a logistic regression model, we find that lymphoid cells are more similar across heart, SKM and kidney, whereas there is less concordance for myeloid cells (Extended Data Fig. 8f, Supplementary Tables 16, 19), probably due to tissue-specific adaptability of myeloid cells[51]. Notably, populations corresponding to cardiac monocyte-derived macrophages, *LYVE1+* MP1, *DOCK4+* MP1–2 and antigen-presenting macrophages are absent from the SKM and kidney. Cardiac *LYVE1+* MP2–3, pro-inflammatory monocytes, classical monocytes and mast cells are more similar to their SKM counterparts, indicating greater similarity of striated muscle and cardiac myeloid populations versus kidney. The transcriptional signature of cardiac *LYVE1+* MP2-3 is specific without overlap in SKM and kidney (Fig. 4c).

## Conduction system and neuronal cells

Among 3,961 cells expressing prototypic electrophysiologic transcripts (*NRXN1*, *NRXN3* and *KCNMB4*), we identify six neuronal cell subclusters (Extended Data Fig. 9a–c). NC1 constitutes 75–80% of neuronal cells and exhibits a basal gene program including *LGI4*, required for glia development and axon myelination[52]. NC2 and NC4 show strong expression of the central nervous system marker, *PRKG1*[53], and co-express typical fibroblast and cardiomyocyte genes, respectively. NC3 has overlapping gene expression signatures with ECs. NC5 expresses *LGR5*, a Wnt signalling, G-protein-coupled receptor and stem-cell marker that promotes cardiomyocyte differentiation in the outflow tract[54], an arrhythmogenic area[55]. This cluster also expresses the neurodegenerative disease gene *PPP2R2B*[56] (verified by smFISH) (Extended Data Fig. 9d, Supplementary Table 20); *LSAMP*, which guides the development of specific patterns of neuronal connections[57]; and the lipoprotein transport enzyme *LPL* that remyelinates damaged neurons[58]. NC6 mimics Schwann cells, expressing *MBP*, *PRX* and *MPZ*, which encode myelin constituents[59].

## Adipocytes

Cardiac adipocytes uniformly express *GPAM*, *FASN* and *ADIPOQ* and at lower levels, *LEP*[60] (Extended Data Fig. 9e–h). ADIP1 expresses genes for PPAR pathways, metabolism of lipids and lipoproteins, and lipolysis. ADIP2 expresses ECM genes such as *LAMA2*, *IGFBP7* and *FBN1*, which encodes both the glycoprotein fibrillin1 and asprosin, a white adipose tissue secreted hormone involved in glucose homeostasis (Supplementary Table 21). Given a stromal-related molecular signature,

ADIP2 cells may represent fibrogenic adipocytes and/or precursors[61,62]. ADIP3 transcripts encode inflammatory and cytokine responsive molecules.

## COVID-19 and GWAS disease relevance

Transcripts encoding the severe acute respiratory syndrome coronavirus 2 (SARS-CoV-2) receptor *ACE2*[63] are highest in pericytes, followed by FBs and lowest in cardiomyocytes, where expression is twofold higher in ventricular than atrial cardiomyocytes. Among proteases priming viral entry[63], *TMPRSS2* transcripts are absent in pericytes, FBs and cardiomyocytes, whereas *CTSB* and *CTSL* are lowly expressed with higher levels in cardiomyocytes. *ACE2* expression in pericytes and fibroblasts is depicted by smFISH (Extended Data Fig. 10a–e).

We define cells enriched for genes from 12 cardiovascular genome-wide association studies (GWAS) and involved in SARS-CoV-2 infection using MAGMA[64] (Extended Data Fig. 10f). Atrial fibrillation GWAS signals are associated with transcriptional profiles in vCM3, owing to higher mean expression of *CAV1*, *CAV2* and *PRRX1*. PR interval GWAS signals are associated with vCM3 and aCM5, with high expression of *SCN5A*, *CAV1*, *ARHGAP24*, *MEIS1*, *TBX5* and *TTN*. GWAS signals for QRS duration are associated with specific gene expression in NC2 (*PRKCA*, *CEP85L*, *SLC35F1*, *SIPA1L1*, *KLF12* and *FADS2*). Coronary artery disease and hypertension GWAS signals are associated with transcripts from many cell lineages, particularly SMCs, FBs, and ECs, reflecting the relevance of vascular cells in both disorders.

## Discussion

Our analyses of approximately half a million single cells and nuclei from six distinct cardiac regions from fourteen donors considerably expand an emerging reference adult heart cell atlas. By combining single-cell and single-nuclear RNA-seq data with machine learning and in situ imaging techniques, we provide detailed insights across the repertoire of cardiac cells, including cardiomyocytes (excluded by single-cell RNA-seq) and ECs (underrepresented in cardiac snRNA-seq). We quantify the cellular composition highlighting chamber-specific features and differences between male and female donors. Within each cell compartment, we identify and validate prototypic lineage-specific genes, and genes with previously unknown cardiac expression. Our results begin to unravel the molecular underpinnings of cardiac physiology and the cellular response to stress and disease.

Cardiomyocytes are the most prevalent cardiac cells and comprise higher percentages in ventricles than atria, and in female versus male ventricular tissues. Transcriptional differences between atrial and ventricular cardiomyocyte populations indicate different developmental origins, distinctive haemodynamic forces and specialized functions in cardiac chambers. Cellular diversity of FBs reveals ECM-producing and ECM-organizing activities that with other cells support cardiomyocytes across varying biophysical stimuli. The vascular compartment contains several ECs and pericyte populations and two SMC subtypes with distinct anatomical and arterio-vascular characteristics. Arterial and venous ECs are predicted to interact with mural cells via Notch signalling pathways involved in regulating vascular homeostasis and development. Immune cells interact with FBs and cardiomyocytes. In addition to confirming previous findings[65,66], we show macrophage complexity and infer paracrine circuits for cardiac homeostasis. Cross-tissue analyses delineate cardiac populations distinct from skeletal muscle and kidney.

We illustrate the relevance of cardiac cell atlas by defining cell lineages enriched in cardiovascular GWAS and molecules involved in SARS-CoV-2 infection. High expression of the viral receptor *ACE2* in pericytes and its correlation with *AGTR1* is consistent with the role of renin–angiotensin–aldosterone system signalling in cardiac haemodynamics[67].

We recognize limitations associated with cell capture by different data sources and unintended bias from surgical sampling. However, we expect our results will inform studies of other cardiac regions (valves, papillary muscle and conduction system), propel studies with large cohorts to determine the roles of age, gender and ancestry on normal cardiac physiology and provide crucial insights to enable mechanistic understanding of heart disease.

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

## Methods

### Data reporting

No statistical methods were used to predetermine sample size. The experiments were not randomized, and investigators were not blinded to allocation during experiments and outcome assessment.

### Research ethics for donor tissues

Heart tissues (donors D1–D7 and D11) were processed at Wellcome Sanger Institute (Hinxton, UK) and obtained from deceased transplant organ donors after Research Ethics Committee approval (ref 15/EE/0152, East of England Cambridge South Research Ethics Committee) and informed consent from the donor families. Heart tissues (donors H2–H7) were processed at Harvard Medical School (Boston, Massachusetts, USA) and obtained from deceased organ donors after Human Research Ethics Board approval Pro00011739 (University of Alberta, Edmonton, Canada). Informed consent from donor families was acquired via the institutional Human Organ Procurement and Exchange Program (HOPE). Cardiovascular history was unremarkable for all donors (Supplementary Table 1).

### Tissue acquisition and processing

Tissues were acquired from UK and North American donors (D1–7 and 11, H2–7) after circulatory death (DCD) (D2, D4–D7 and D11) and after brain death (DBD) (D1, D3, H2–H7). For UK DCD donors, after a five-minute stand-off and for DBD, the chest is opened, the aorta is cross-clamped and cardiac samples are acquired. For North American DBD donors, the aorta is cross-clamped, cold cardioplegia (Celsior) is administered under pressure via the aorta to arrest beating, the heart is excised, rinsed in cold saline and samples acquired. All donor samples were full-thickness myocardial biopsies from the left and right atrium, left and right ventricles, interventricular septum and apex, with intentional exclusion of large epicardial fat deposits. Samples used for single nuclei isolation were flash-frozen and stored at −80 °C. Single-cell isolation and CD45+ enrichment was carried out on freshly collected samples. Residual tissue after nuclei and cell isolation procedures was formalin-fixed or frozen in OCT for additional studies.

All tissues were stored and transported on ice at all times until freezing or tissue dissociation to minimise any transcriptional degradation. Previous studies on the post-mortem tissue stability of the GTEx consortium on bulk tissues[68] and in single cells[69] suggest only minor changes in tissues within the first 24 h post mortem when stored in cold conditions.

### Single nuclei isolation

Single nuclei were obtained from flash-frozen tissues using mechanical homogenization as previously described[70]. Tissues were homogenized using a 7 ml glass Dounce tissue grinder set (Merck) with 8–10 strokes of a loose pestle (A) and 8–10 strokes of a tight pestle (B) in homogenization buffer (250 mM sucrose, 25 mM KCl, 5 mM MgCl$_2$, 10 mM Tris-HCl, 1 mM dithiothreitol (DTT), 1× protease inhibitor, 0.4 U μl$^{-1}$ RNaseIn, 0.2 U μl$^{-1}$ SUPERaseIn, 0.1% Triton X-100 in nuclease-free water). Homogenate was filtered through a 40-μm cell strainer (Corning). After centrifugation (500$g$, 5 min, 4 °C) the supernatant was removed and the pellet was resuspended in storage buffer (1× PBS, 4% bovine serum albumin (BSA), 0.2 U μl$^{-1}$ Protector RNaseIn). Nuclei were stained with NucBlue Live ReadyProbes Reagents (ThermoFisher) and Hoechst-positive single nuclei were purified by fluorescent activated cell sorting (FACS) using influx, XDP or FACSAria (BD Biosciences) (Supplementary Fig. 1). Nuclei purification and integrity was verified under a microscope, and nuclei were further processed using the Chromium Controller (10X Genomics) according to the manufacturer's protocol.

### Single-cell preparation

Heart tissues (0.2–0.9 g) were transferred from cardioplegic solution into gentleMACS C-tubes (Miltenyi Biotec) containing enzymatic digestion base solution (100 μg ml$^{-1}$ liberase TH Research grade and 50 μg ml$^{-1}$ DNase I, HBSS 10 mM HEPES and 30 mM taurine)[71]. Tissues were minced using scissors (FST) and automatically digested using gentleMACS Octo Dissociator (Miltenyi Biotec) with heaters. Cardiomyocyte-depleted single-cell suspension were washed with base solution containing 20% fetal bovine serum (FBS) (Gibco), filtered through 70-μm nylon strainer (BD Falcon), collected by centrifugation (330$g$, 10 min, 4 °C) and resuspended in base solution containing 0.2% FBS (Gibco). Cells were manually counted three times by Trypan blue exclusion after each centrifugation and resuspended at a concentration of at least $2 \times 10^6$ ml$^{-1}$. Single cells were processed using Chromium Controller (10X Genomics) according to the manufacturer's protocol.

### CD45+ cell enrichment

Cell suspension was prepared as described above and subsequently labelled using anti-human CD45 monoclonal antibody-conjugated microbeads according to the manufacturer's protocol (Miltenyi Biotec). In brief, up to $10^7$ cells were incubated for 15 min at 4 °C in 80 μl of PBS, BSA, EDTA buffer (1× PBS pH 7.2, 0.5% BSA, 2 mM EDTA) containing 20 μl CD45 microbeads. Cell suspension was washed in PBS, BSA, EDTA buffer once and collected by centrifugation (330$g$, 10 min, 4 °C). Resuspended cells were applied to MACS LS columns (Miltenyi Biotec). CD45-depleted cell fraction was discarded after three washes with PBS, BSA and EDTA buffer and the CD45+ cell fraction was collected in PBS, BSA and EDTA buffer by removal of the columns from the magnetic field. CD45+ cells were counted and resuspended in PBS, BSA and EDTA buffer to a concentration of at least $2 \times 10^6$ per ml before further processing using a Chromium Controller (10X Genomics) according to the manufacturer's protocol.

### Chromium 10X library preparation

Single cells and nuclei were manually counted by Trypan blue exclusion or automatically using a Countess II (Life Technologies) using at least two separate counts. Cell or nuclei suspension was adjusted to 400–1,000 cells per microlitre and loaded on the Chromium Controller (10X Genomics) with a targeted cell or nuclei recovery of 4,000–10,000 per reaction. 3′ gene expression libraries were prepared according to the manufacturer's instructions of the v2 or v3 Chromium Single Cell Reagent Kits (10X Genomics). Quality control of cDNA and final libraries was done using Bioanalyzer High Sensitivity DNA Analysis (Agilent) or 4200 TapeStation System (Agilent). Libraries were sequenced using HiSeq 4000 (Illumina) at Wellcome Sanger Institute, and NextSeq 500 (Illumina) at Harvard Medical School with a minimum depth of 20,000–30,000 read pairs per cell or nucleus (Supplementary Table 22).

### Spatial validation using smFISH with RNAscope probes

During preparation of formalin-fixed paraffin-embedded (FFPE) samples fresh tissue was fixed in neutral-buffered 10% formalin for 18–36 h and subsequently embedded in paraffin blocks. Fixed-frozen tissue samples were fixed in 4% paraformaldehyde (ThermoFisher). Sections were cut at 5-μm thickness using a microtome and placed onto Super-Frost Plus slides (VWR). FFPE tissue slides were automatically stained using BOND RX (Leica) and the RNAscope Multiplex Fluorescent Reagent Kit v2 Assay (ACDBio) according to the manufacturer's protocol. Fixed-frozen tissue slides were processed according to the protocol of RNAscope Multiplex Fluorescent Assay v1 (ACDBio). RNAscope ready- or custom-made target probes were run in parallel to multiplex positive and negative controls (Extended Data Fig. 12b, Supplementary Table 23). All nuclei were DAPI-stained. All FFPE tissue slides were imaged using an Opera Phenix High-Content confocal Screening System (Perkin Elmer) with a 1-μm $z$-step size and 20× water-immersion objective (NA 0.16, 0.299 μm per pixel). Channels: DAPI (excitation 375 nm, emission 435–480 nm), Atto 425 (excitation 425 nm, emission 463–501 nm), opal 520 (excitation 488 nm, emission 500–550 nm), opal 570 (excitation 561 nm, emission 570–630 nm), opal 650 (excitation

640 nm, emission 650–760 nm). Fixed-frozen tissue slides were imaged using a LSM710 confocal microscope (Zeiss) and 40× oil-immersion objective (1.3 oil, DIC III). Channels: DAPI (excitation 375 nm, emission 435–480 nm), Alexa Fluor 488 (excitation 492 nm, emission 517 nm), Atto 550 (excitation 560 nm, emission 575 nm) and Atto 647 (excitation 649 nm, emission 662 nm). Visualization and background removal (rolling ball radius) were done using Fiji/ImageJ[72]. Pseudocolours were used for better visualization.

## Haematoxylin and eosin staining

Tissue samples were fresh-frozen in isopentane (ThermoFisher) at −80 °C and embedded in OCT (VWR). Sections were cut at a thickness of 10 μm using a microtome, placed onto SuperFrostPlus slides (VWR) and further processed according to a standard haematoxylin and eosin staining protocol (Extended Data Fig. 12a).

## Acquisition of skeletal muscle tissue

Intercostal muscle samples were obtained from between the second and third rib on the left side. This is typically from the deepest layer of muscle (furthest away from the skin). Samples were collected directly into the cold preservation solution.

## Nuclei isolation for skeletal muscle

Muscle tissue was washed in 1× PBS, cleaned of any visible fat depositions and minced to obtain fragments of approximately 1 mm$^3$. Per sample, approximately 0.3 g of minced tissues was homogenized in 3 ml of buffer A (250 mM sucrose, 10 mg ml$^{-1}$ BSA, 5 mM MgCl$_2$, 0.12 U μl$^{-1}$ RNaseIn, 0.06 U μl$^{-1}$ SUPERasIn, 1× protease inhibitor) using Dounce tissue grinder set (Merck) with 50 strokes of the loose pestle (A). The homogenate was filtered through a 100-μm cell strainer (Corning) and the strainer was washed with 1 ml and then 750 μl of buffer A. After the addition of Triton X-100 (final concentration 0.5%), the mixture was further homogenized with 50 strokes of the tight pestle (B). After filtering through a 40-μm strainer, nuclei were centrifuged (3,000g, 5 min, 4 °C), resuspended in 1 ml of buffer B (320 mM sucrose, 10 mg ml$^{-1}$ BSA, 3 mM CaCl$_2$, 2 mM magnesium acetate, 0.1 mM EDTA, 10 mM Tris-HCl, 1 mM DTT, 1× protease inhibitor, 0.12 U μl$^{-1}$ RNaseIn, 0.06 U μl$^{-1}$ SUPERasin) and purified using a 27% Percoll gradient solution. The Percoll mixture was centrifuged at 20,000g (15 min, 4 °C) and the pellet was resuspended in 200 μl of buffer B, followed by centrifugation (20,000g, 3 min, 4 °C). After Trypan Blue staining, the intact nuclei were counted using a haemocytometer. Nuclei were profiled using a Chromium Controller (10X Genomics) according to the manufacturer's protocol.

## Single-cell isolation for skeletal muscle

Muscle tissue was washed in 1× PBS, cleaned of any visible fat depositions and finely minced. Then, 2 g of the minced tissue was transferred to digestion buffer 1 (750 U ml$^{-1}$ collagenase type 2 in 1× PBS) and incubated at 37 °C in a water bath for 90 min. The partially digested tissue was collected by centrifugation (650g, 5 min, 4 °C) and the pellet was resuspended in digestion buffer 2 (100 U ml$^{-1}$ collagenase type 2, 2 U ml$^{-1}$ dispase in PBS). After 30 min incubation at 37 °C in a water bath, the digestion was stopped by the addition of 2% FBS. Cells were filtered through a 100-μm and a 40-μm nylon strainer (BD Falcon), collected by centrifugation (650g, 4 °C, 3 min) and washed with 1× PBS, 2% FBS. Subsequently, a 20% Percoll gradient (15,000g, 4 °C, 20 min) was used for cell purification. The layer containing cells was collected, washed in PBS containing 2% FBS, and viable cells were counted by Trypan Blue exclusion using a haemocytometer. Nuclei were profiled using a Chromium Controller (10X Genomics) according to the manufacturer's protocol.

The methods key resources table is in Supplementary Table 24.

## Transcriptome mapping

After sequencing, samples were demultiplexed and stored as CRAM files. Each sample was mapped to the human reference genome (GRCh38 v.3.0.0) provided by 10X Genomics, and using the CellRanger suite (v.3.0.1) with default parameters. Single-cell samples were mapped against the reference as it was provided. Single-nuclei samples, the reference for pre-mRNA, was created using the 10X Genomics instructions (https://support.10xgenomics.com/single-cell-gene-expression/ software/pipelines/latest/advanced/references).

## Count data processing

After mapping, samples from each data source (single nuclei, single cell and CD45$^+$ cell) were grouped into individual AnnData objects by concatenating the raw_feature_bc_matrix.h5.h5 and adding the appropriate metadata information. For each data source object, the mean of unique molecular identifiers (UMIs) (n_counts) was calculated and used as a threshold for empty droplets.

## Doublet detection

After removal of empty droplets, we applied scrublet[73] to assign a doublet score (scrublet_score) to each cell. These cells were clustered and visualized using the UMAP method[74]. In addition, each cell was processed for doublet detection using a percolation method to allow for improved detection of doublets[75].

## Cell quality control and filtering

Each data source was processed and annotated separately to account for source-specific quality differences. These metrics are included as covariates for further processing. Total cells and CD45$^+$ cells were filtered for counts (500 < n_counts <15,000), genes (200 < n_genes), mitochondrial genes (percent_mito <20%), ribosomal genes (percent_ribo <20%) and scrublet score (scrublet_score <0.3). Single nuclei were filtered for counts (500 < n_counts <15,000), genes (300 < n_genes <6,000), mitochondrial genes (percent_mito <5%), ribosomal genes (percent_ribo <5%) and scrublet score (scrublet_score <0.3). The same filtering thresholds were applied to the skeletal muscle dataset.

Scanpy toolkit 1.5[76] in Python v.3.7 was used to perform downstream analyses, including normalization (normalize_per_cell: counts_per_cell_after = 10,000), log transformation (log1p), variable gene detection (highly_variable_genes), regressing out unwanted sources of variation (regress_out: n_counts and percent_mito), data feature scaling (scale: max_value = 10) and PCA (pca: using highly variable genes) as previously described[77].

## Batch alignment using deep variational autoencoder

We built a global manifold by aligning all the data sources and donors in our data. This was done in a three-step procedure: (1) Each source was analysed and annotated separately, aligning only for donors using a pericyte-space linear regression step before batch alignment with bbknn[78]. Differentially expressed genes (DEGs) were calculated using a Wilcoxon rank sum test with Bonferroni–Hochberg adjustment as implemented in the Scanpy framework. (2) To annotate each cluster, we used an integrative approach by searching the top significant DEGs ($P < 1 × 10^{-5}$) with a logFC >1 against the ToppFun[79] and EnrichR[80] databases. Significant hits on pathways, transcriptional regulation and biological processes were prioritized to annotate a given cluster. Each cellular compartment was labelled under the adata.obs['cell_type'] slot after grouping source-specific cell states. (3) All sources were combined into a single AnnData object under the label adata.obs['cell_sources']. Batches were aligned using the batch_correction function from the scGen variational autoencoder[81]. First we align for adata.obs['cell_sources'], using adata.obs['cell_type'] as an anchor. Next, we aligned for adata.obs['donor'], using adata.obs['cell_type'] as an anchor. Each batch alignment round was run for 50 epochs.

Manifolds for the adipocytes, vascular and immune cardiac populations, as well as the skeletal muscle analysis, were created using this method and the clustering accuracy was evaluated with SCCAF[82] (Extended Data Fig. 12d).

## DEGs

To help with the annotation of the subpopulations of each cell compartment, we calculated the DEGs using the Wilcoxon rank sum test as implemented in the scanpy workflow and recommended by recent benchmarking studies[83]. A gene was considered to be differentially expressed if it has a $\log_2$-transformed fold change >1 and a $P < 1 \times 10^{-5}$, unless stated otherwise in the analysis section.

## Cell–cell interactions

Expression matrices of the populations under study were exported from the AnnData, together with a metadata table that contained the cell-barcodes as indices. We then ran CellPhoneDB as follows: cellphonedb method statistical_analysis meta.tsv counts.tsv–counts-data = gene_name–threads = 60. CellPhoneDB raw predictions were filtered by removing those interactions with a $P > 1.0 \times 10^{-5}$. Significant pairs were then submitted for gene set enrichment analysis into ReactomeDB, enrichR and ToppFun for functional classification. The vascular cells were randomly sub-sampled to 39,000 cells before the analysis, and the cardiac repair group (atrial and ventricular cardiomyocytes, FBs, and immune cells) was randomly sub-sampled to 69,295 cells before the analysis.

## Visualization of gene expression on 10X Genomics Visium data

We processed the publicly available left ventricular myocardium Visium data from 10X Genomics (https://support.10xgenomics.com/spatial-gene-expression/datasets/1.0.0/V1_Human_Heart) using the Scanpy v.1.5 workflow adapted for the analysis of 10X Genomics Visium data (https://scanpy-tutorials.readthedocs.io/en/latest/spatial/basic-analysis.html). In brief, spots were removed with less than 500 UMIs or more than 20,000 UMIs, and less than 200 genes. Data were log-transformed and normalized before plotting.

## Estimation of RNA velocity

To calculate the RNA velocity of the single cells and CD45+ enriched single cells, we used the CellRanger output BAM file and the GENCODE v33 GTF (ftp://ftp.ebi.ac.uk/pub/databases/gencode/Gencode_human/release_33/gencode.v33.chr_patch_hapl_scaff.annotation.gtf.gz) file together with the velocyto[84] CLI v.0.17.17 to generate a loom file containing the quantification of spliced and unspliced RNA. Next, we built a manifold, cluster the cells and visualize the RNA velocities using scVelo[85].

## Subpopulation analyses of atrial and ventricular cardiomyocytes, FBs and neuronal cells

All barcodes labelled in the global object as cardiomyocytes, fibroblasts and neural cells were selected for further subpopulation analyses. Additional cell population-specific filtering criteria were applied to nuclei as follows: cardiomyocyte counts (n_counts <12,500), genes (n_genes <4,000), mitochondrial genes (percent_mito <1%), ribosomal genes (percent_ribo <1%) and scrublet score (scrublet_score <0.25); FB mitochondrial genes (percent_mito <1%), ribosomal genes (percent_ribo <1%); neuronal cell genes (n_genes <4000), mitochondrial genes (percent_mito <1%), ribosomal genes (percent_ribo <1%). Total and CD45+ cells were excluded in the atrial and ventricular cardiomyocytes datasets and did not contribute to subpopulation analysis. No further filtering of FBs or neuronal cell total and CD45+ cells was applied. Cardiomyocytes and FBs were then further split into two groupings based on the region of origin: (1) left and right atrium, and (2) left and right ventricles, apex and interventricular septum.

Donor effects were aligned as described in step (1) above. For FB and neuronal cells, sources were aligned as described in step (3) above. Leiden clustering and UMAP visualization were performed for identifying subpopulations and visualization[86]. Differentially expressed genes were calculated using the Wilcoxon rank sum test. Genes were ranked by score.

## Cross-tissue comparison of cardiac immune populations with skeletal muscle, kidney and blood immune populations

We collected single-cell transcriptome data for adult kidneys from ref. [99] (https://www.kidneycellatlas.org/), and subset all immune cells reported in their study. For the SKM we selected the annotated immune cells from the merged manifold. For the human blood, we used the publicly available 10,000 single PBMC cells dataset provided by 10X Genomics (https://support.10xgenomics.com/single-cell-gene-expression/datasets/3.0.0/pbmc_10k_v3). As previously described[87], we trained a logistic regression model on the cardiac immune cells using 80% of the expression data and tested its accuracy on the remaining 20% to produce a model with an accuracy of 0.6862 (Extended Data Fig. 8f, Supplementary Table 16). We then applied this model to predict analogue cardiac immune populations in the adult kidney, SKM and PBMCs. Predictions with a probability less than 0.8 were excluded from downstream comparative analyses.

## Gene Ontology enrichment analysis

For the ventricular cardiomyocyte population, we used the R package gProfileR (https://cran.r-project.org/web/packages/gProfileR/index.html) with the score-ranked gene list of vCM4 as input and the set of genes expressed in ventricular cardiomyocytes as background (those genes having a UMI count >1). To perform the Gene Ontology analysis on the vascular cells, the top 500 significant DEGs ($P < 1 \times 10^{-5}$) with a log-transformed fold change >1 were searched against the Gene Ontology biological process database using ToppFun[79] (Supplementary Table 25). The top five significantly enriched terms ($q < 0.05$) for each subpopulation were selected and plotted on a heat map. To perform the pathway analysis on the adipocytes, the top 500 significant DEGs ($P < 1 \times 10^{-5}$) with a log-transformed fold change > 0.5 were searched against ToppFun[79] pathway databases (Supplementary Table 26). The top five significantly enriched pathways ($q < 0.05$) for each subpopulation were selected and plotted on a heat map.

## Gene set score

We use the score_genes function as implemented in scanpy to calculate the enrichment of genes involved in the Oncostatin M pathway. A list of genes was collected upon literature research[88,89]. For gene set enrichment, only highly expressed genes were considered to reduce noise (more than 500 UMIs across all cells). The same analysis was performed for comparison of cardiac immune cells in our study with the observations of previous studies on cardiac-resident macrophages[51], mouse tissue-remodelling macrophage[45] and yolk sac lineage origin[90].

## Statistics and reproducibility

All analyses were performed using R Software, v.3.6.1. Student's $t$-tests were used to compare cell type distributions at each site. $P < 0.05$ was considered statistically significant. Linear regression models (correlations) were obtained using the R linear model function (lm), which estimates statistical likelihood ($P$ value) of a linear relationship. Bonferroni correction was applied for multiple testing.

The depicted RNAscope micrographs in the figures are representative. The micrographs in Figs. 2g, 3c, h and Extended Data Fig. 3c (*HAMP*), Extended Data Fig. 3e (*CNN1*), Extended Data Figs. 4g, m, 6f were repeated with similar results in two individual tissue sections. The micrographs in Figs. 2h, 3f and Extended Data Fig. 3c (*CNN1*), Extended Data Fig. 3e (*PCDH7*), Extended Data Figs. 6e, h, 9d were repeated with similar results in three individual tissue sections. The micrographs in Figs. 1e, 2d, 3e and Extended Data Figs. 1f, 3c (*FHL1*) and Extended Data Fig. 6d were repeated with similar results in four individual tissue sections. The micrographs in Fig. 2c and Extended Data Fig. 3c (*PRELID2*), Extended Data Figs. 10e, 12a were repeated with similar results in six or more individual tissue sections. Positive and negative controls were done once per used samples.

## GWAS enrichment analysis

We downloaded GWAS summary statistics from broad cvdi, EBI GWAS catalogue and GWAS atlas. We selected traits with well-powered GWAS ($n > 5,000$ and number of significant loci >10). GWAS datasets are summarized in Supplementary Table 27. Gene expression data of protein-coding genes were mapped onto Entrez gene ids and these gene annotations were used on the human genome assembly hg19/37. We only used gene expression data from nuclei. We implemented the analysis previously described[64] in python and in R. The log-transformed counts (plus one pseudocount) were used to compute average cell type-specific expression profiles. We performed individual magma analyses for each cell type, always conditioning on default gene level covariates (for example, gene length) and average gene expression across all cells. Subsequently, we applied the Benjamini–Hochberg method and selected cell type trait associations with FDR < 10%. These pairs were then subjected to conditional analysis as previously described[64] to define 'independent', 'jointly explained' and 'partially jointly explained' pairs of associations (Supplementary Table 28).

## Distributions of dispersed cells and isolated nuclei

The different procedures for obtaining isolated nuclei and dispersed cells resulted in significantly different distributions of cell types (Supplementary Table 29, Extended Data Fig. 2). Notably, 30.1% and 49.2% of isolated nuclei were derived from atrial and ventricular cardiomyocytes in the atrial and ventricular regions, whereas these cells were mostly excluded from preparations of isolated and CD45-selected cells (Supplementary Table 2).

Excluding cardiomyocytes, the distribution of cell types identified from isolated nuclei and dispersed cells remained distinct (Supplementary Table 30). Although 59.0% of dispersed cells were ECs, only 15.7% of nuclei were derived from ECs. By contrast, 64.2% of nuclei were from FBs (31.2%) and pericytes (33.0%), whereas only 17.1% of dispersed cells were FBs (2.3%) and pericytes (14.8%). These differences may reflect sensitivity of EC nuclei to isolation procedures or resistance of pericytes and FBs to cellular enzymatic digestion.

Despite differences in cell distributions between isolated nuclei and dispersed cells, the gene expression profiles of cell lineages were reasonably correlated ($r > 0.4$ for each cell type). To address the concordance of the genes captured by cells and nuclei, we compared the expression of the major cell type markers from Fig. 1c across the three sources (Extended Data Fig. 1c). As nuclei lack cytoplasmic RNA, the expression of certain genes, especially immune genes *NKG7* and *C1QA*, was lower in nuclei than in cells. Nevertheless, the general trend with respect to marker genes was consistent across the three sources, and the same genes distinguished individual cell types independent of the source.

## Further analysis of vascular cells

The PC3_str contained similar contribution of cells and nuclei, and had a scrublet score below the stringent threshold used; nevertheless, the average number of genes and counts in this cluster was higher than average. Thus, despite our stringent quality filtering, we cannot exclude the possibility that there might be doublets in this cluster. EC10_CMC-like and PC4_CMC-like co-express EC or pericyte genes with cardiomyocyte markers and further studies are required to understand whether they represent previously unknown cell states or doublets.

The observations of the arterial and venous SMC are supported by previous studies, which predict that arterial SMCs are more contractile, and venous SMCs are less differentiated[91].

EC3_cap enrich for transcripts encoding components of AP1 (*JUN* and *FOS*), which mediates multiple EC fate decisions including response to VEGF, inflammatory and stress signals, and *ATF3*, an adaptive-response gene induced by diverse signals[92–94].

## Skeletal muscle characterization

We collected intercostal skeletal muscle samples from five healthy individuals, including one donor with matched cardiac tissue, and profiled the transcriptome of 35,665 single cells and 39,597 single nuclei. Analogous to the heart, the combination of cells and nuclei allowed us to capture and resolve major cell lineages, including cardiomyocyte, fibroblasts, endothelial cells, smooth muscle cells, pericytes, myeloid and lymphoid immune cells and satellite cells (Extended Data Fig. 11a, b, Supplementary Table 31).

Further analysis of the vascular cells of the skeletal muscle identified ten distinct populations. The endothelial cells showed five clusters separated based on their respective vascular beds with signatures similar to the ones we observe in the heart. The EC_cap expresses *VWF* and *RGCC*. Venous EC_ven express *ACKR1* and *PLVAP*, whereas arterial EC_art show *SEMA3G* and *HEY1*, in line with our heart data (Extended Data Fig. 11c, d, Supplementary Table 32).

The overall distributions of vascular and stromal cell populations in skeletal and cardiac muscle were similar, including the arterial and venous features of ECs; however; skeletal muscle contained a single SMC cluster, potentially related to the smaller size of the dataset. In skeletal muscle, the predicted cell–cell interactions of the EC_art and SMCs included NOTCH1/4–JAG1 as well as JAG1/JAG2/DLL4–NOTCH3, but not JAG1/JAG2/DLL4–NOTCH2, inferred in the heart (Extended Data Fig. 11e, f, Supplementary Table 33).

## Cardiac immune cells

Using the logistic regression model, we did not identify any counterpart of the cardiac *IL17RA*+ monocytes in SKM or kidney, possibly owing to the small size of this population.

Naive T cells (CD4+T_naive) identified expressed *CCR7* and *SELL*, indicative of their naive and tissue-resident nature[95]. Memory T cells (CD8+T_tem) expressed *BACH2*, *STAT4* and *IL7R*, associated with long-term immune memory[96,97]. We further characterized the lymphoid cells using scNym[98], and trained it using published data[99,100]. The resulting model was applied to our cardiac immune cells and those cells, with a predicted score higher than 0.8 were presumed to be likely candidates for re-annotation. Using this approach, we identified candidates for plasma B cells (109), dendritic cells (645), innate lymphoid cells (89), MAIT T cells (219), T helper cells (80) T regulatory cells (11), T central memory cells (103), γδ T cells (30) and plasmocytoid dendritic cells (27). These annotations can be found in the cardiac immune object annotations under the label 'scNym' at www.heartcellatlas.org.

## Reporting summary

Further information on research design is available in the Nature Research Reporting Summary linked to this paper.

## Data availability

Data objects with the raw counts matrices and annotation are available via the www.heartcellatlas.org webportal. Raw data are available through the Human Cell Atlas (HCA) Data Coordination Platform (DCP) with accession number: ERP123138 (https://www.ebi.ac.uk/ena/browser/view/ERP123138). The 10X Genomics Visium data for the heart left ventricle tissue can be accessed at: https://support.10xgenomics.com/spatial-gene-expression/datasets/1.1.0/V1_Human_Heart. GWAS data used in this study can be found in Supplementary Table 27. All of our data can be explored at www.heartcellatlas.org.

## Code availability

All code used for this study can be accessed as Jupyter notebooks in the project GitHub repository: https://github.com/cartal/HCA_Heart.

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

**Acknowledgements** This publication is part of the Human Cell Atlas (www.humancellatlas.org/publications). We acknowledge the Sanger Flow Cytometry Facility, Sanger Cellular Generation and Phenotyping (CGaP) Core Facility, and Sanger Core Sequencing pipeline for support with sample processing and sequencing library preparation. We acknowledge the HMS MicRoN Core for their support and assistance in this work. We thank J. Eliasova for graphical images; M. Prete, V. Kiselev and O. Tarkowska for IT support; S. Aldridge for editing the manuscript; C. Domínguez-Conde, J. Park, K. James, K. Tuong, R. Elmentaite and M. Haniffa for discussions about immune cell annotation, and M. Luecken, M. Lotfollahi, D. Fischer and F. Theis for discussions on computational analyses. We thank T. R. W. Oliver and L. Campos for their assistance with orientation of basic cardiac histology. M.N. is the recipient of a British Heart Foundation (BHF) grant (PG/16/47/32156) and J.J.B. is recipient of a BHF grant (FS/13/12/30037, PG/15/57/31580, PG/17/71/33242). M.N., N.H. and S.A.T. are funded by a BHF/DZHK grant (SP/19/1/34461). N.H. is the recipient of an ERC Advanced Grant under the European Union Horizon 2020 Research and Innovation Program (grant agreement AdG788970) and the Federal Ministry of Education and Research of Germany in the framework of CaRNAtion (031L0075A). N.H., C.E.S. and J.G.S. are supported by grants from the Leducq Fondation (16CVD03). D.R. is supported by the German Research Foundation (DFG). H.Z. is supported by the long-term Chinese Council Scholarship (CSC). M.K. received funding provided by the Alexander von Humboldt Foundation. H.Z. was supported by an EMBO long-term postdoctoral fellowship. G.O. received funding provided by the CIHR Canadian Institutes for Health Research, HSF Heart and Stroke Foundation and AI Alberta Innovates. This project has been made possible in part by grant number 2019-202666 from the Chan Zuckerberg Initiative (C.E.S., J.G.S., M.N., M.H., N.H. and S.A.T.). C.T.-L., K.P., E.F., E.T., K.R., O.B., H.Z. and S.A.T. are supported by the Wellcome Sanger Institute grant (WT206194) and the Wellcome Science Strategic Support for a Pilot for the Human Cell Atlas (WT211276/Z/18/Z)). J.G.S. and C.E.S. are supported by NIH grants (2R01HL080494, 2R01HL080494, 1UM1HL098166). J.G.S. and C.E.S. are supported by Engineering Research Centers Program of the National Science Foundation under NSF Cooperative Agreement no. EEC-1647837. C.E.S. and B.M. are supported by the Howard Hughes Medical Institute. We are grateful to the deceased donors and their families, the Human Organ Procurement and Exchange Program (HOPE) and to the Cambridge Biorepository for Translational Medicine (CBTM) for access to human tissue. E.R.N. was supported by a fellowship from the Sarnoff Cardiovascular Research Foundation.

**Author contributions** Conceived the study: N.H., S.A.T., M.N., C.E.S., J.G.S., H.M. Acquired tissue: K.M., K.S.P., G.O., H.Z., A.V. Processed tissue: M. Litviňuková, D.R., S.S., E.F., L.T., H.W., J.M.G., B.M., D.M.D., K.R., M.N. Computational methods development: C.T.-L., K.P., M.H., J.G.S. Cell biology methods development: M. Litviňuková, E.L.L., M.N., H.M., D.R. Histopathological evaluation: J.J.B., M.N. Data processing: C.T.-L., K.P., G.P. Global analysis: C.T.-L., D.R., M. Litviňuková, C.L.W., E.L.L., J.G.S. Cardiomyocyte analysis: D.R., C.L.W., E.R.N., H.M., D.M.D., N.H., J.G.S., C.E.S. Vascular analysis: M. Litviňuková, M.N., C.T.-L., M.K., S.A.T. Fibroblast analysis: E.L.L., H.M., D.R., M. Litviňuková, M.K., M.N., J.G.S., C.E.S., N.H. Immune analysis: C.T.-L., S.A.T. Neuronal analysis: E.R.N., D.R., C.L.W., J.G.S., C.E.S. Adipocyte analysis: M. Lee, M.N. Comparative analysis: C.T.-L., M. Litviňuková, M.N., S.A.T. GWAS analysis: M.H., C.T.-L., N.H., S.A.T. Interpreted results: M. Litviňuková, C.T.-L., D.R., H.M., C.L.W, E.L.L., M.H., J.B., M.N., C.E.S., J.G.S., S.A.T., N.H. Wrote the manuscript: M. Litviňuková, C.T.-L., H.M., D.R., C.L.W., E.L.L., M.K., M.H., J.G.S., C.E.S., M.N., N.H., S.A.T.

**Competing interests** The authors declare no competing interests.

**Additional information**
**Correspondence and requests for materials** should be addressed to J.G.S., C.E.S., M.N., N.H. or S.A.T.

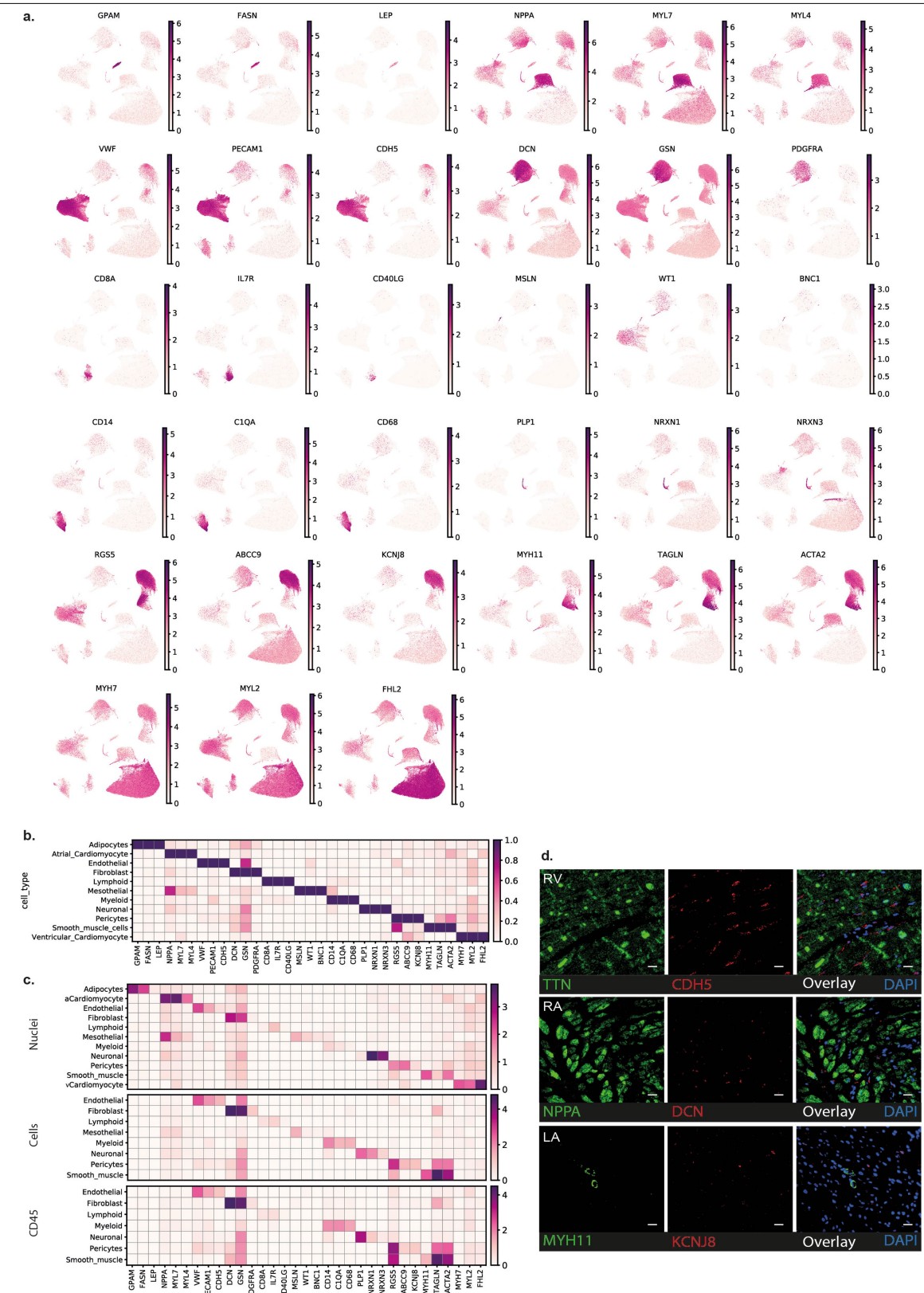

**Extended Data Fig. 1 | Expression of the canonical markers. a**, UMAP embedding of selected canonical markers shown in Fig. 1c. **b**, Scaled expression ($\log_2$-transformed fold change, $\log_2$FC) of selected canonical markers shown in Fig. 1c. **c**, Expression ($\log_2$FC) of marker genes from Fig. 1c in each source highlighting that the same marker genes are used for identification of the same cell types in both cells and nuclei. **d**, Multiplexed smFISH staining of cell type-specific transcripts from Fig. 1e in right ventricles (top): *TTN* (green, cardiomyocytes) and *CDH5* (red, EC) right atrium (middle): *NPPA* (green, aCM) and *DCN* (red, FB) and LA (bottom): *MYH11* (green, SMC) and *KCNJ8* (red, pericytes), nuclei are DAPI-stained (dark blue). Scale bars, 20 μm. For details on statistics and reproducibility, see Methods.

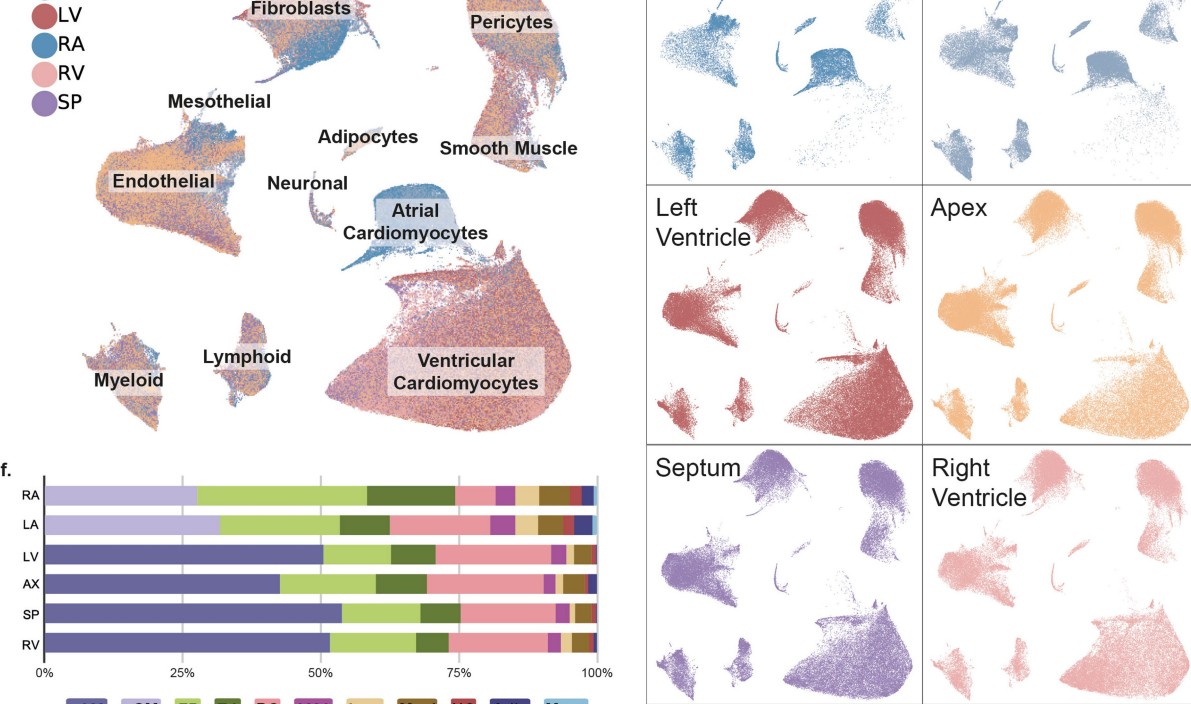

**Extended Data Fig. 2 | Source and region covariates of the global dataset. a**, UMAP embedding of the major cell types coloured by source. **b**, UMAP embedding highlighting the individual sources **c**, Distribution of cell types obtained by each source. Data are available in Supplementary Table 29. Further analyses and descriptions are available in the Methods and Supplementary

Table 30. **d**, UMAP embedding of the major cell types coloured by region. **e**, UMAP embedding highlighting the individual regions **f**, Distribution of cell types across the six sampled regions (nuclei only). Data are available in Supplementary Table 2.

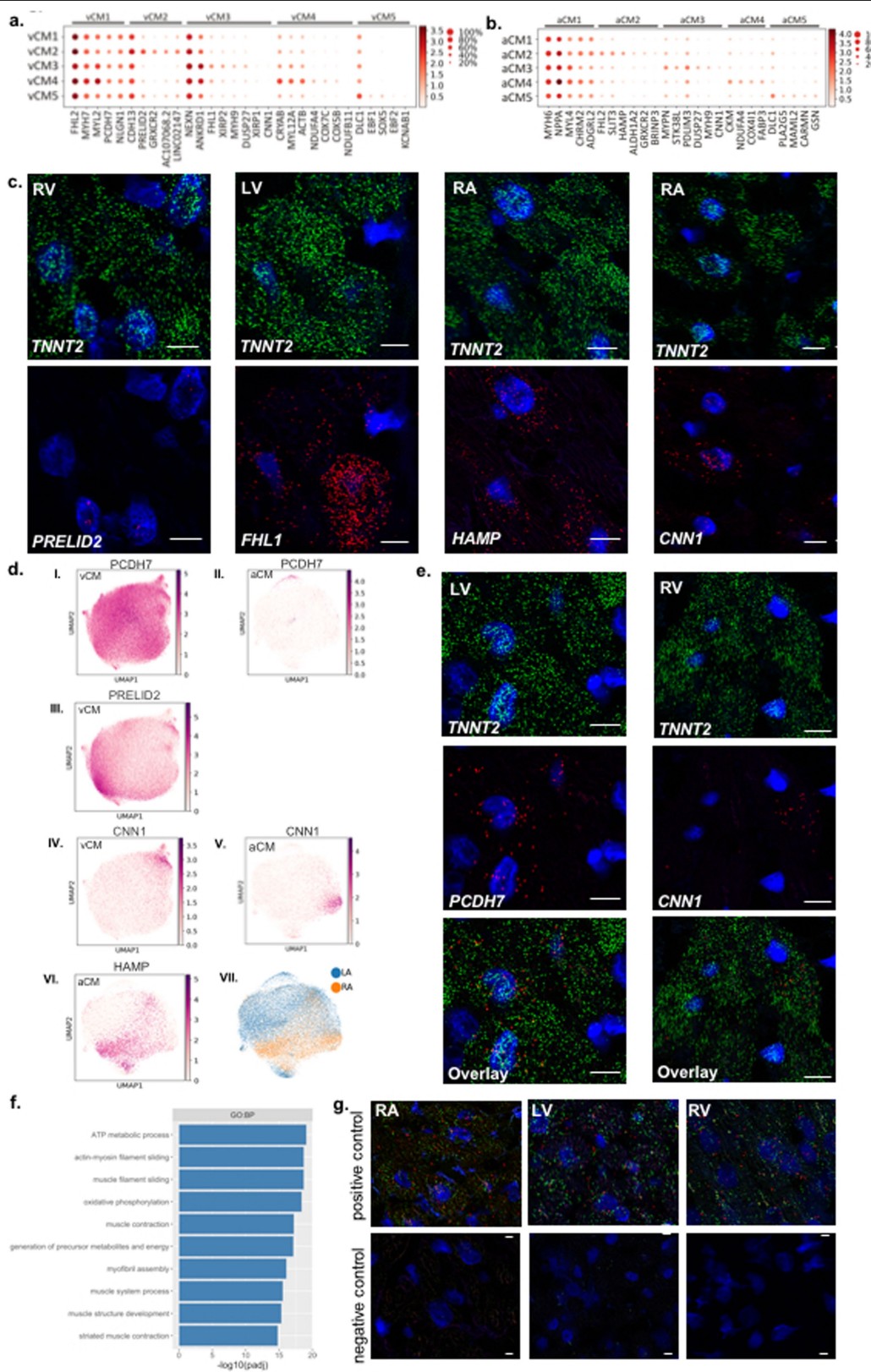

**Extended Data Fig. 3** | See next page for caption.

**Extended Data Fig. 3 | Ventricular and atrial cardiomyocytes. a**, Expression (log$_2$FC) of selected marker genes in ventricular cardiomyocyte subpopulations. **b**, Expression (log$_2$FC) of selected marker genes in atrial cardiomyocyte subpopulations **c**, Single channel multiplexed smFISH images of overlay shown in Fig. 2c, d, g, h. **d**, Expression (log$_2$FC) of specific markers in cardiomyocyte subpopulations. I and II, *PCDH7* expression in ventricular and atrial cardiomyocytes, respectively. III, *PRELID2* expression is highest in vCM2 and is enriched in right ventricles. IV and V, *CNN1* expression is enriched in both vCM3 and aCM3. VI and VII, *HAMP* expression is enriched in the right atrium.

**e**, Multiplexed smFISH of transcripts enriched in cardiomyocyte subpopulations. Left, expression of *TNNT2* (green) and *PCDH7* (red) in left ventricles. Right, expression of *TNNT2* (green) and *CNN1* (red) in right ventricles, nuclei are DAPI-stained (dark blue). Scale bars, 10 µm. **f**, Gene Ontology analysis results for vCM4 showing significant terms related to energy metabolism and muscle contraction. Data are available in Supplementary Table 6. **g**, Multiplexed smFISH of positive and negative RNAscope control probes. Scale bars, 5 µm. For details on statistics and reproducibility, see Methods.

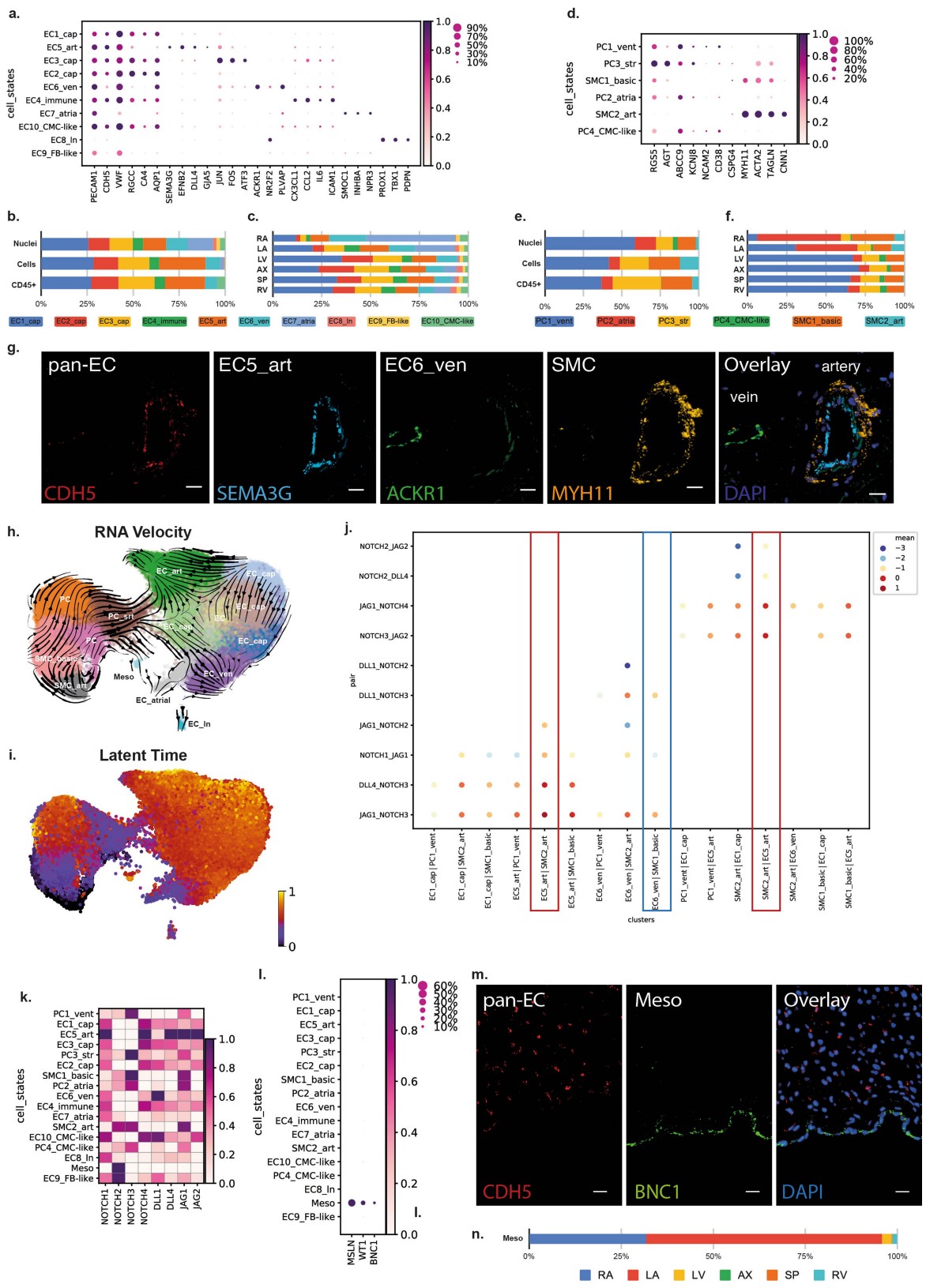

**Extended Data Fig. 4** | See next page for caption.

**Extended Data Fig. 4 | Vascular and mesothelial populations. a**, Scaled expression (log₂FC) of selected marker genes for EC subpopulations. **b**, **c**, Distribution of the EC subpopulations across the sources (**b**) and the regions (**c**) (nuclei only). Data are available in Supplementary Table 9. **d**, Scaled expression (log₂FC) of selected marker genes of pericytes and smooth muscle cell subpopulations. **e**, **f**, Distribution of the mural subpopulations across the sources (**e**) and the regions (**f**) (nuclei only). Data are available in Supplementary Table 9. **g**, Multiplexed smFISH in apex section shows *MYH11* (yellow) expression in vascular SMC (thick in artery and very thin in nearby small calibre vein), *CDH5* (red) in the endothelium, and *SEMA3G* (cyan) and *ACKR1* (green) expression respectively in arterial and venous ECs, nuclei are DAPI-stained (dark blue). Scale bars, 20 µm. **h**, UMAP embedding of vascular and mesothelial cells with stochastic representation of the RNA velocity. **i**, Latent time of the vascular cells showing predicted directionalities of the cell populations based on the RNA splicing dynamics. The analysis uses only cells, nuclei are omitted. EC_cap, capillary ECs; EC_art, arterial ECs; EC_ven, venous ECs; EC atrial, atrial endothelial cells; EC_ln, lymphatic endothelial cells; PC, pericytes; PC_str, stromal pericytes; SMC_basic, smooth muscle cells; SMC_art, arterial smooth muscle cells. **j**, Predicted cell–cell interactions using the CellphoneDB statistical inference framework on 39,000 cells from 14 biologically independent individuals (*n* = 14). Selected ligand–receptor interactions show specificity of NOTCH ligands-receptors pairing in defined vasculature beds. Mean of combined gene expression of interacting pairs (log₂FC). CellPhoneDB *P* value of the specificity of the interactions = 10 × 10−5. The red rectangles highlight the arterial interactions and the blue rectangle highlights venous interactions depicted in Fig. 3d. Notably, even though the EC6_ven and SMC2_art interaction is unexpected, we cannot exclude that those cell states are restricted to their respective vascular beds. Further validation is needed to determine the exact spatial distribution of EC6_ven and SMC2_art and subsequently verify whether the interaction is plausible in vivo. Data are available in Supplementary Table 10. **k**, Scaled expression (log₂FC) of the ligands and receptors from **g** across the vascular populations described in Fig. 3a. **l**, Scaled expression (log₂FC) of selected marker genes of mesothelial cells. **m**, Multiplexed smFISH localizes the mesothelial cells expressing *BNC1* into the epicardium of the left atria. *CDH5* shows endothelial cells in the tissue and is absent from the mesothelial cells, nuclei are DAPI-stained (dark blue). Scale bars, 20 µm. **n**, Distribution of the mural subpopulations across the sampled regions (nuclei only). Data are available in Supplementary Table 9. For details on statistics and reproducibility, see Methods.

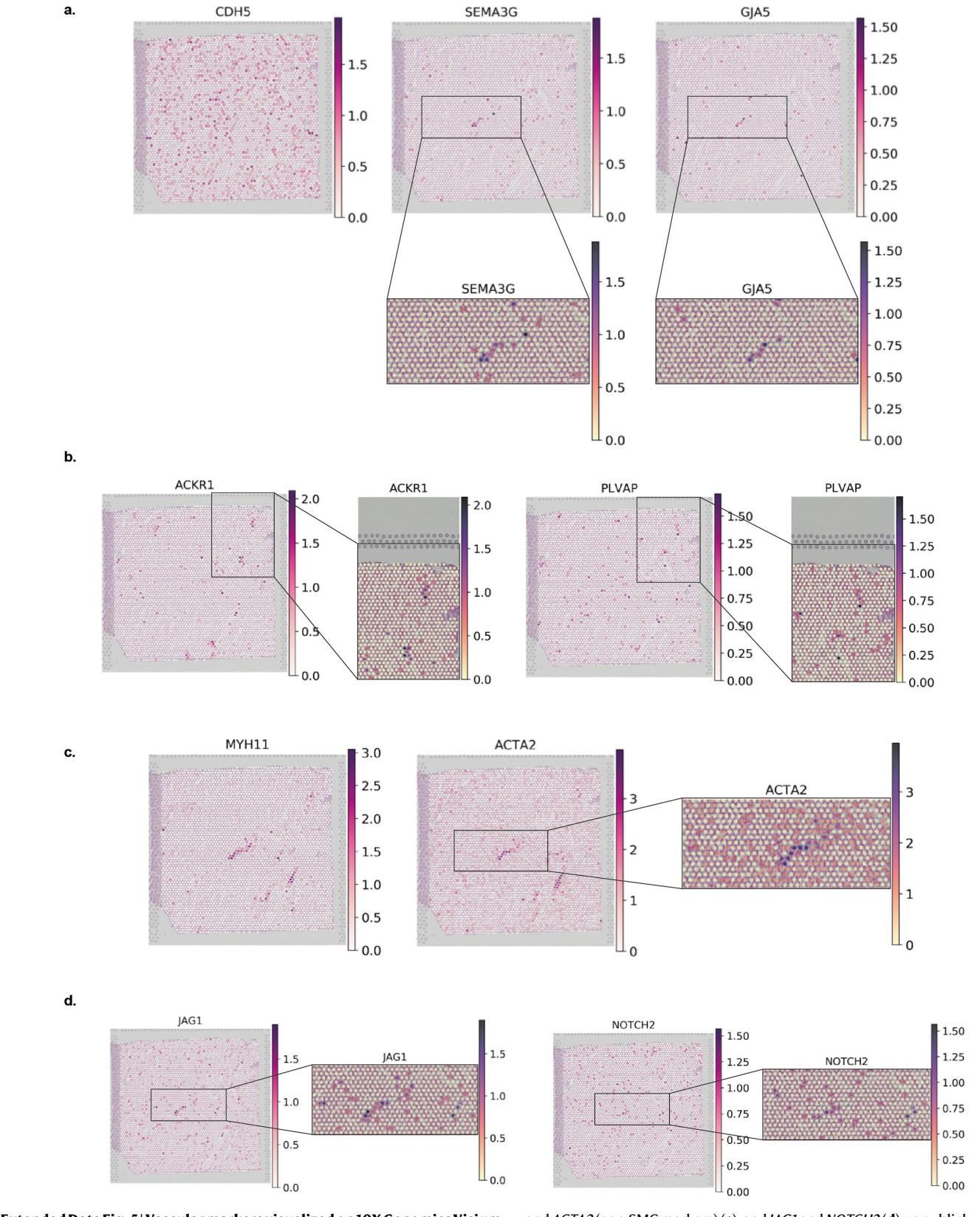

**Extended Data Fig. 5 | Vascular markers visualized on 10X Genomics Visium data. a**–**d**, Spatial expression (log₂FC) of *CDH5* (pan-EC marker), *SEMA3G* and *GJA5* (arterial EC markers) (**a**), *ACKR1* and *PLVAP* (venous EC markers) (**b**), *MYH11* and *ACTA2* (pan-SMC markers) (**c**), and *JAG1* and *NOTCH2* (**d**) on publicly available 10X Visium section of human left ventricle. *JAG1* and *NOTCH2* are the predicted interaction partners for arterial ECs and SMCs, respectively.

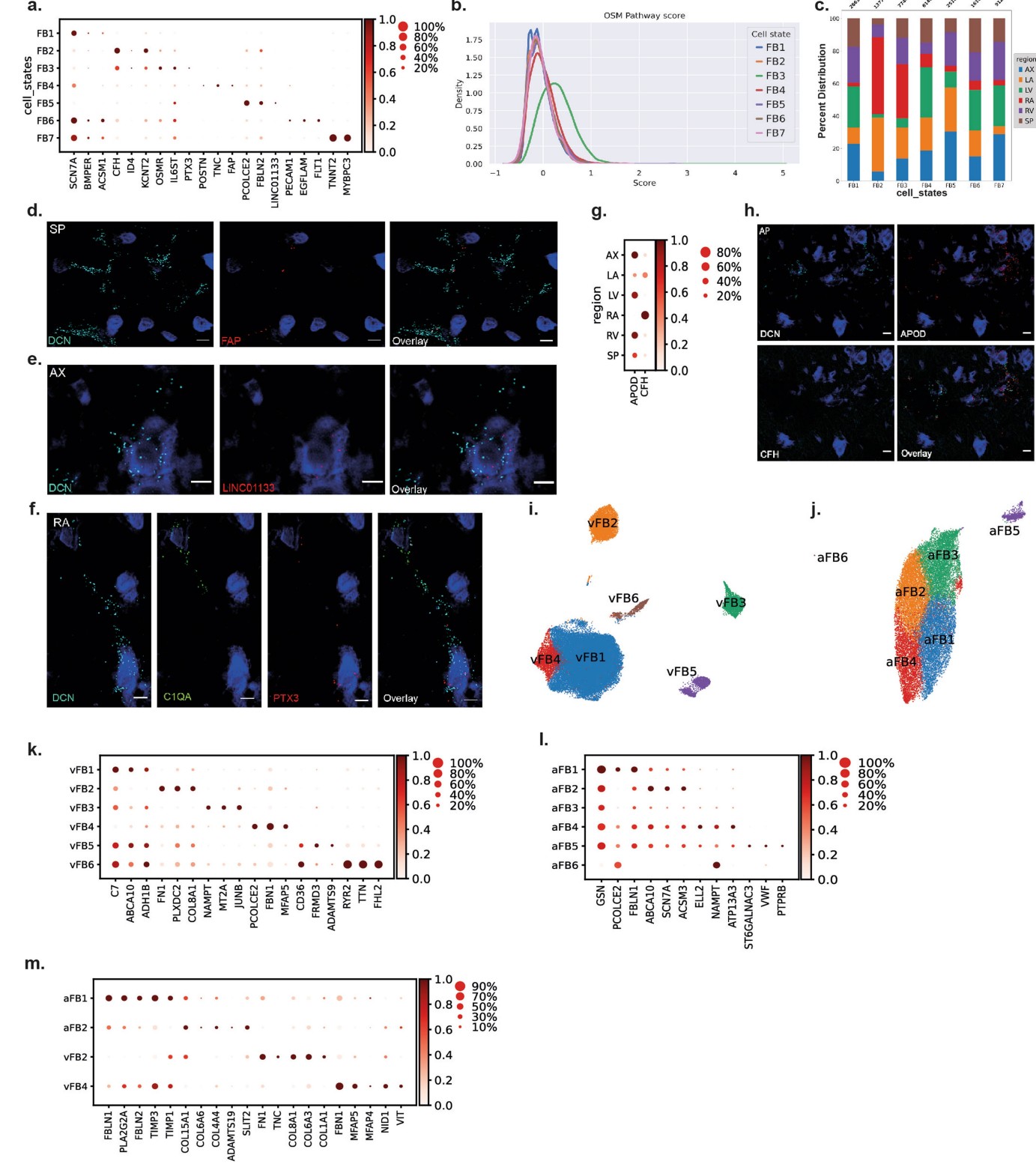

**Extended Data Fig. 6** | See next page for caption.

**Extended Data Fig. 6 | Fibroblasts. a**, Scaled expression ($\log_2$FC) of selected marker genes of identified FB populations. **b**, Enrichment for oncostatin M pathway for FB populations showing enriched activity in FB3. A list of genes with which the score was calculated is in Supplementary Table 12. **c**, Regional distribution per FB population. Some FB populations show enrichment in the atria (left and right), such as FB2 and FB3. FB1, FB4–FB6 are enriched in the ventricles (left, right, apex and interventricular septum). Data are available in Supplementary Table 35. **d**–**f**, Multiplexed smFISH for probes targeting *FAP*, *LINC01133* and *PTX3* confirming FB4, FB5 and FB3 subpopulations. *FAP* (red) is imaged in interventricular septum, *LINC01133* (red) in apex and *PTX3* (red) in right atrium tissue section. *DCN* (cyan) is used as a pan-FB marker, *C1QA* (green) as a pan-macrophage marker, nuclei are DAPI-stained (dark blue). Scale bars, 5 µm. **g**. Scaled expression ($\log_2$FC) of *APOD* and *CFH* genes, which represent differences between ventricular and atrial fibroblasts. **h**, Multiplexed smFISH of apex section representing *DCN* (cyan), *APOD* (red) and *CFH* (green), nuclei are DAPI-stained (dark blue). Although the APOD signal colocalized with DCN, expression of CFH was absent. Scale bars, 5 µm. **i**, UMAP embedding of the ventricular fibroblast cell-states. **j**, UMAP embedding of atrial fibroblasts cell types. **k**, Scaled expression ($\log_2$FC) of marker genes for ventricular fibroblast subpopulations. **l**, Scaled expression ($\log_2$FC) of marker genes for atrial fibroblast subpopulations. **m**, Scaled expression ($\log_2$FC) of ECM genes differentiating atrial (aFB1, aFB2) and ventricular (vFB2, vFB4) clusters which suggest different ECM mechanisms. For details on statistics and reproducibility, see Methods.

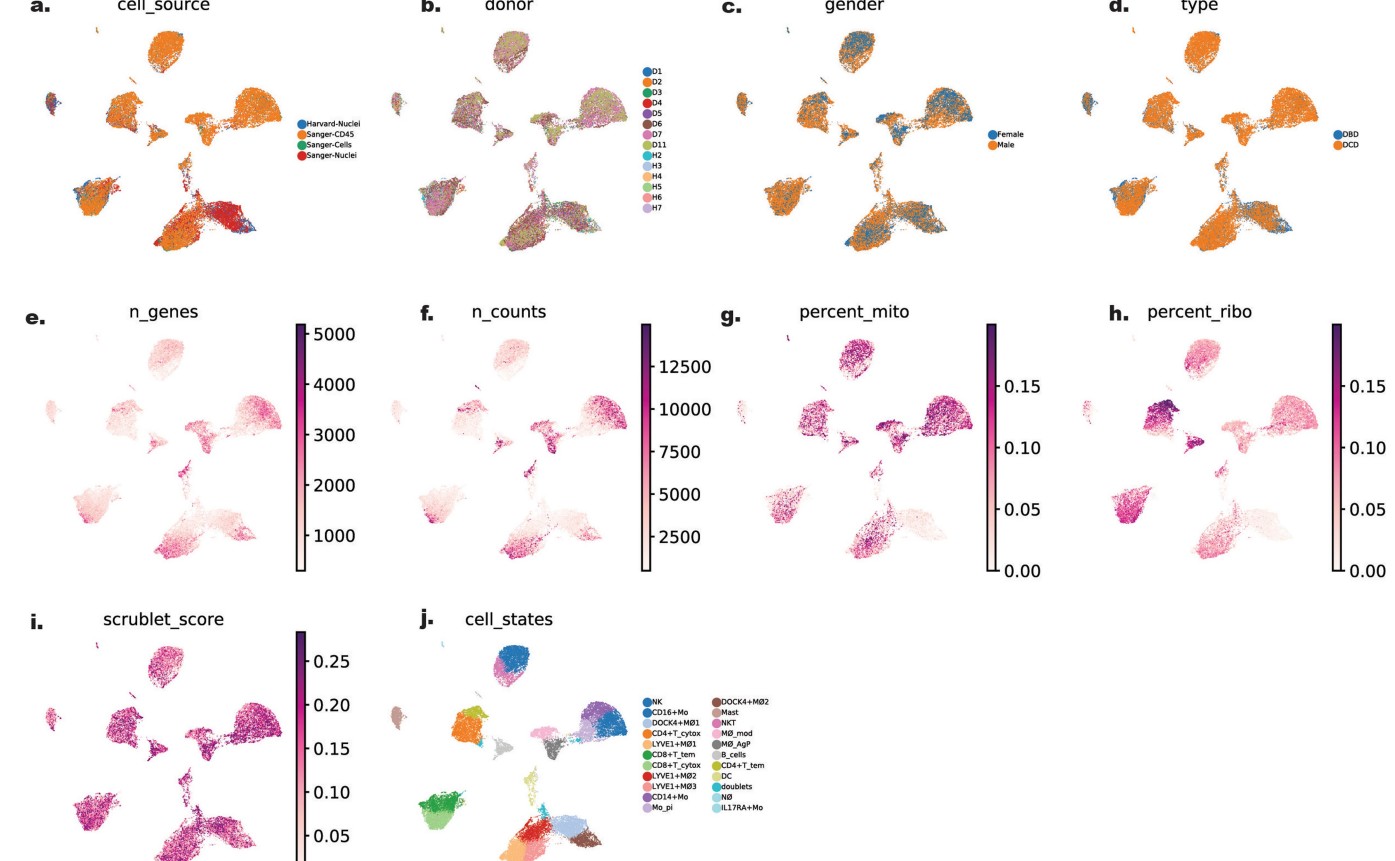

**Extended Data Fig. 7 | Covariates of immune cardiac populations.**
**a–j**, UMAP embedding of cell source (**a**), donor (**b**), gender (**c**), type (**d**), number of genes (**e**), number of counts (**f**), percentage of mitochondrial genes (**g**), percentage of ribosomal genes (**h**), scrublet score (**i**) and annotation of the cell populations of the immune cells (**j**).

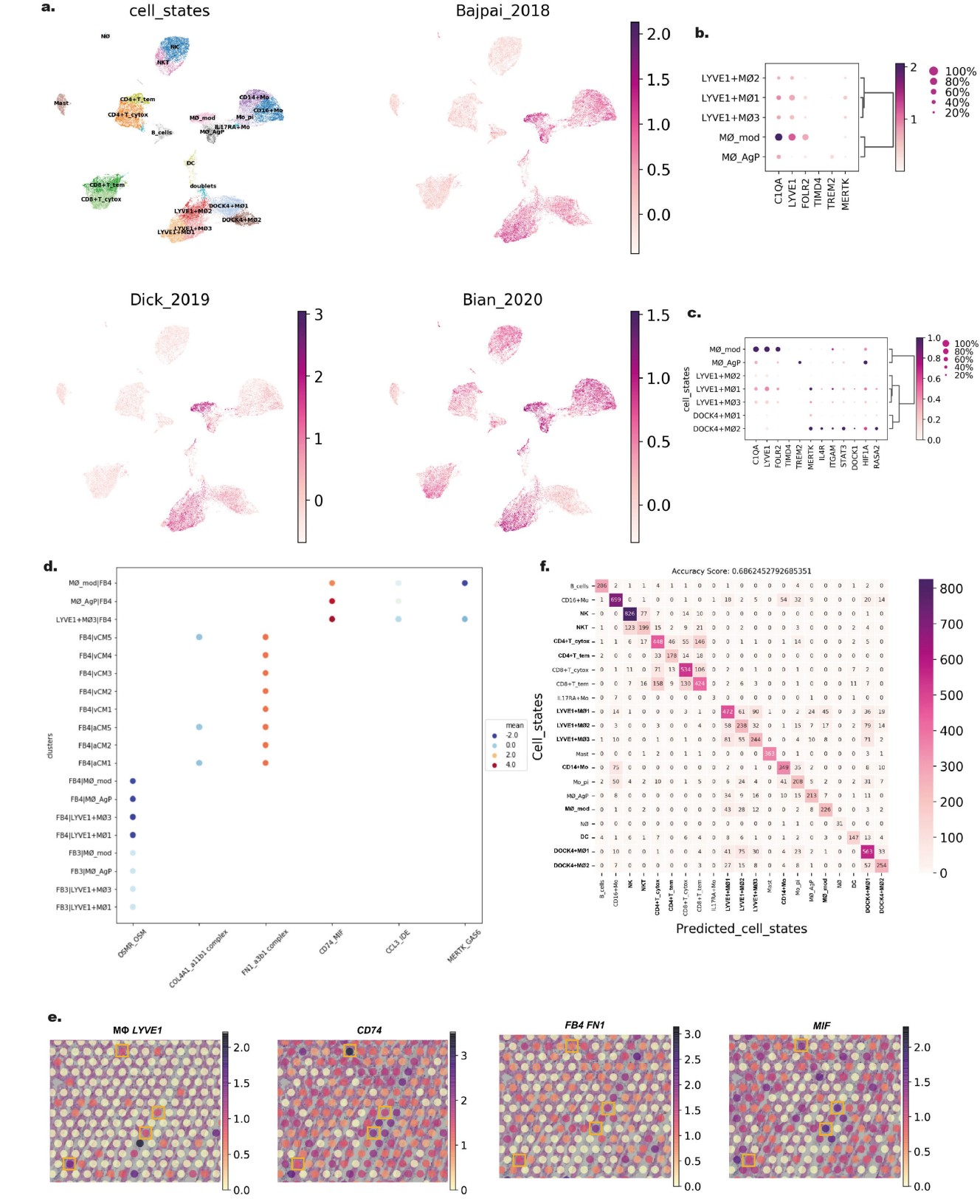

**Extended Data Fig. 8** | See next page for caption.

**Extended Data Fig. 8 | Immune cardiac populations. a**, Visualization of transcriptional signatures from published studies. The score values represent the likelihood of the external transcriptional signature to be present when comparing it against the transcriptional background of a cardiac immune population. Bajpai_2018 = *CCR2-MERTK*⁺ tissue-resident macrophages from ref. [51]. Dick_2019 = self-renewing tissue macrophages from ref. [45]. Bian_2020 = yolk sac-derived macrophages from ref. [90]. The complete signature can be found in Supplementary Table 19. **b**, Expression (log₂FC) of *LYVE1*, *FOLR2* and *TIMD4* characteristic of the self-renewing tissue-resident murine macrophages previously described[45], as well as *MERTK* as previously described[51] and the *TREM2* expression associated to lipid-associated macrophages (LAM) previously described[46]. Complete signatures can be found in Supplementary Table 19. **c**, Scaled expression (log₂FC) of genes differentiating DOCK4⁺ MP1 from DOCK4⁺ MP2: *IL4R*, *ITGAM*, *STAT3*, *DOCK1*, *HIF1A* and *RASA2*. **d**, Predicted cell–cell interactions calculated for 69,295 cardiomyocytes, fibroblasts and myeloid cells from 14 donors (*n* = 14) and enriched for 'extracellular matrix organization'. Mean of combined gene expression of interacting pairs (log₂FC). Data are available in Supplementary Table 17. **e**, Spatial mapping of the *CD74–MIF* interaction between LYVE1⁺MP and FB4 on a publicly available 10X Genomics Visium dataset for left ventricular myocardium. We identified four spots where we observe co-expression of *FN1*, *LYVE1*, *CD74* and *MIF*, as predicted from the cell–cell interactions. The bar represents the log₂FC. **f**, Confusion matrix for the logistic regression model trained on cardiac immune cells. This model reached an accuracy score of 0.6862, showing a stronger accuracy with lymphoid cells, compared with the myeloid ones.

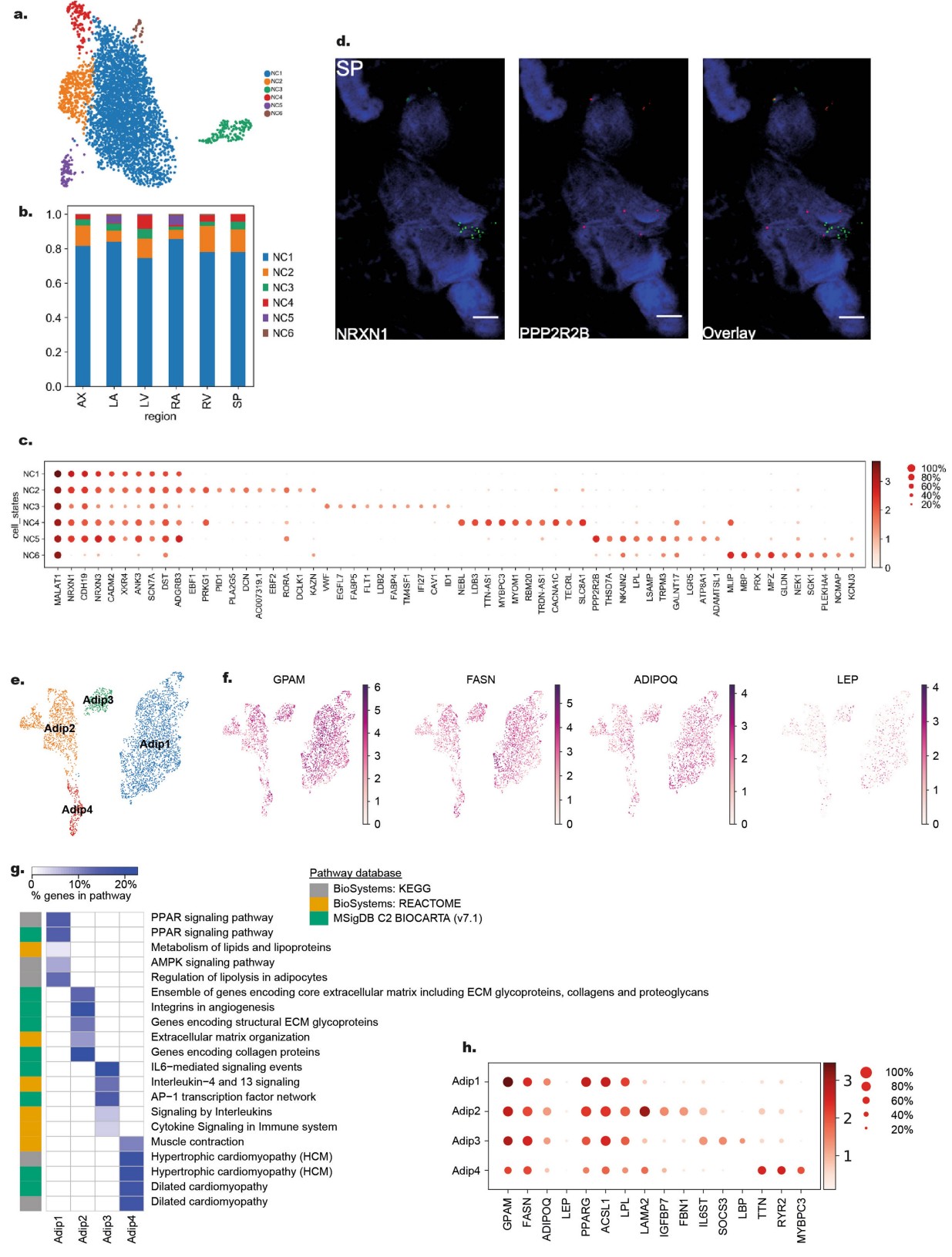

**Extended Data Fig. 9** | See next page for caption.

**Extended Data Fig. 9 | Neuronal and adipocyte populations. a**, UMAP embedding identifies six neuronal subpopulations. **b**, Regional distribution of neuronal cell subpopulations identified in **a**. Data are available in Supplementary Table 20. **c**, Expression (log$_2$FC) dot plot of selected marker genes in neuronal cell subpopulations. **d**, Multiplexed smFISH of *NRXN1* (green) and *PPP2R2B* (red), nuclei were DAPI-stained (dark blue). Scale bars, 5 μm. For details on statistics and reproducibility, see Methods. **e**, UMAP embedding showing four adipocyte subpopulations. **f**, UMAP embedding of expression of gene markers associated with adipocytes (*GPAM*, *FASN*, *ADIPOQ*, *LEP*). **g**, Top five significantly enriched pathways for each adipocyte subpopulation, using differentially expressed genes calculated using the Wilcoxon rank sum test with Benjamini–Hochberg correction (logFC >0.5, $P < 1.0 \times 10^{-5}$) and tested using a hypergeometric distribution with Bonferroni correction as implemented in ToppFun. Data are available in Supplementary Table 21. **h**, Expression (log$_2$FC) of adipocyte associated genes and select marker genes from the top enriched pathway for each adipocyte subpopulation.

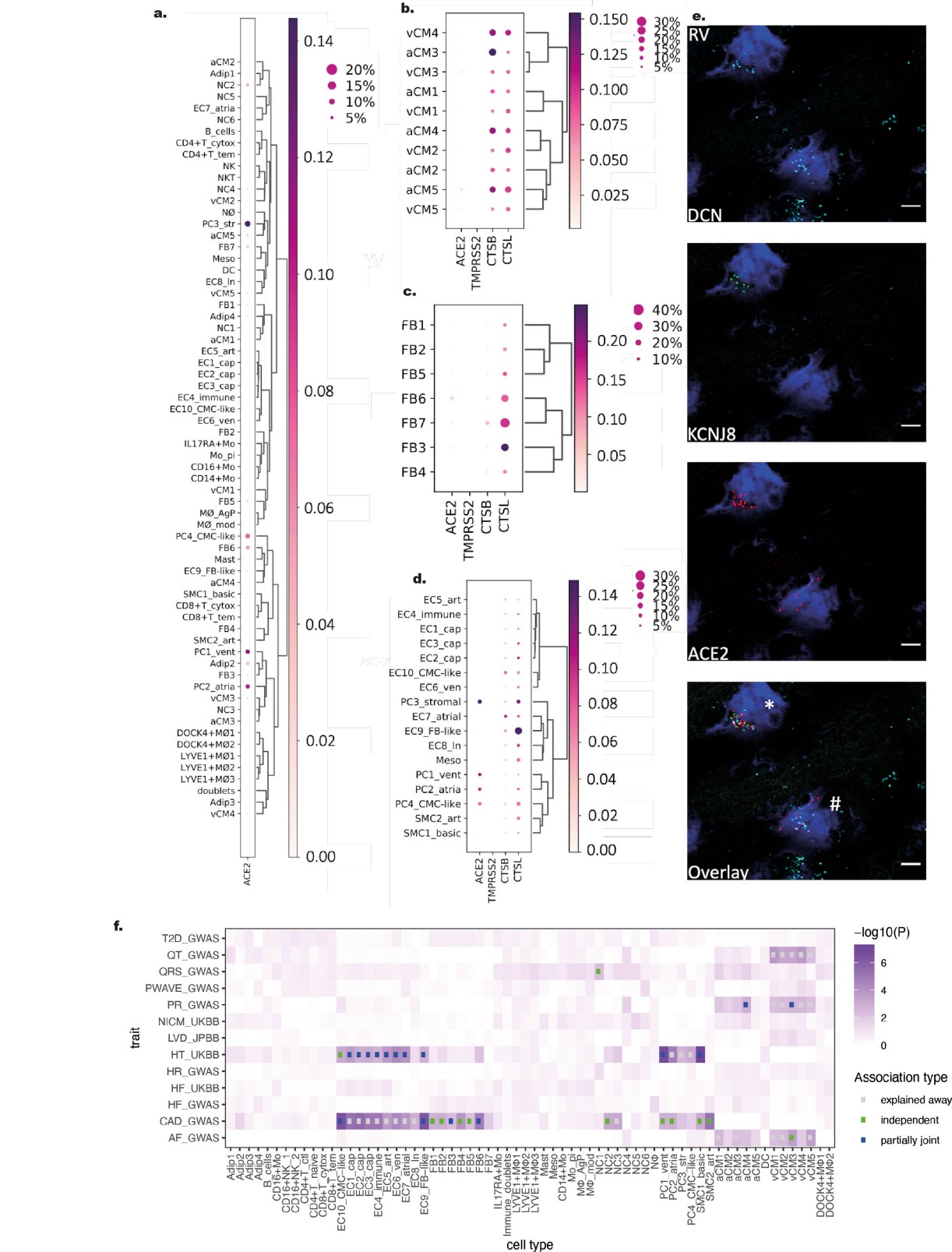

**Extended Data Fig. 10** | See next page for caption.

**Extended Data Fig. 10 | Relevance for COVID-19 and GWAS studies. a**, Global expression (log$_2$FC) of *ACE2* in all cardiac cells. **b**–**d**, Gene expression of *ACE2*, *TMPRSS2*, *CTSB* and *CTSL* in cardiomyocytes (**b**), FBs (**c**) and vascular cells (**d**). **e**, Multiplexed smFISH expression of *DCN* (cyan), *KCNJ8* (green) and *ACE2* (red), nuclei are DAPI-stained (dark blue) marking fibroblasts (#; expression of *DCN*) and pericytes (*; co-expression of *DCN* and *KCNJ8*) in right ventricular tissue section. Scale bars, 5 μm. For statistics and reproducibility, see Methods. **f**, The colour coding of the heat map shows the −log$_{10}$($P$ value) of the MAGMA GWAS enrichment analysis for the association between cell type-specific expression (*y* axis) and GWAS signals (*x* axis). The cell types refer to the subcluster annotations and GWAS studies refer to Supplementary Table 27. AF, atrial fibrillation; CAD, coronary artery disease; HF, heart failure; HR, heart rate; HT, hypertension; LVD, left ventricular diameter; NICM, non-ischaemic cardiomyopathy; PR, PR interval; PWAVE, P-wave duration; T2D, type 2 diabetes; QRS, QRS complex duration; QT, QT interval. Dots mark significant associations (FDR < 10%). The colour of the dots indicates the type of association as determined by pairwise conditional analysis (green: independent association, blue: partially jointly explained with other cell types, grey: explained away by other cell types). Data are available in Supplementary Table 28.

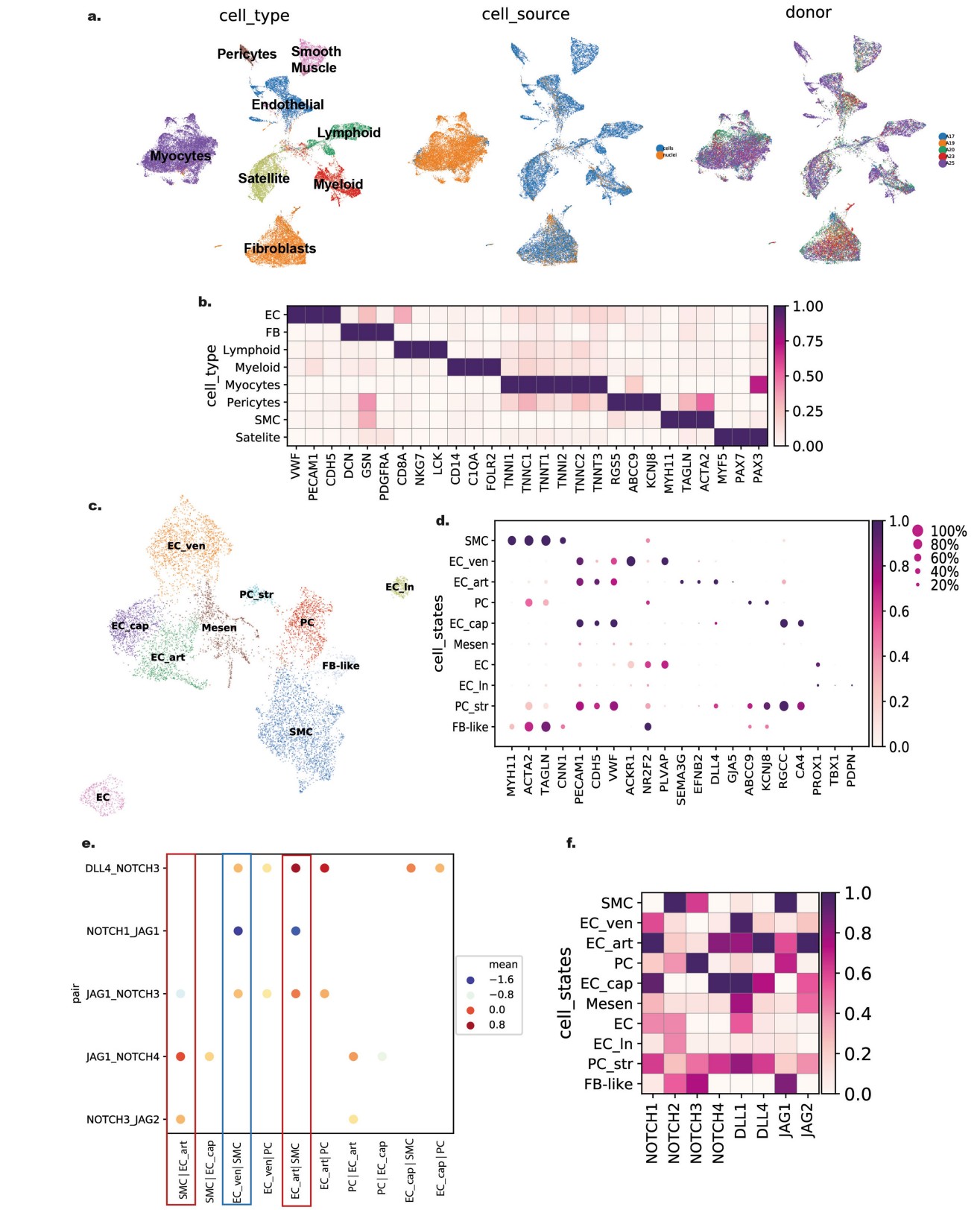

**Extended Data Fig. 11** | See next page for caption.

**Extended Data Fig. 11 | Skeletal muscle populations. a**, UMAP embedding of transcriptional data from skeletal muscle using cells and nuclei. Mural, pericytes and smooth muscle cells. **b**, Scaled expression (log$_2$FC) of selected markers for the major skeletal muscle populations. **c**, UMAP embedding of vascular and stromal populations of skeletal muscle. **d**, Scaled expression (log$_2$FC) of marker genes used in Extended Data Fig. 3 for identification of vascular cell states. **e**, Predicted cell–cell interactions inferred using CellphoneDB statistical inference framework in skeletal muscle cells with 9,220 cells from five donors ($n$ = 5) depicting cell states from **c**, Selected ligand–receptor interactions show specificity of NOTCH ligand–receptor pairing in defined vasculature beds. The interactions of EC_art-SMC are highlighted by a red rectangle and EC_ven-SMC are highlighted by a blue rectangle. Colour of the dots indicates the mean expression level of interacting molecule in partner 1 and interacting molecule partner 2. Mean of combined gene expression of interacting pairs (log$_2$FC). CellPhoneDB $P$ value of the specificity of the interactions = $10 \times 10^{-5}$. Data are available in Supplementary Table 10. **f**, Scaled expression (log$_2$FC) of the ligands and receptors from Extended Data Fig. 3 depicted on vascular populations of skeletal muscle.

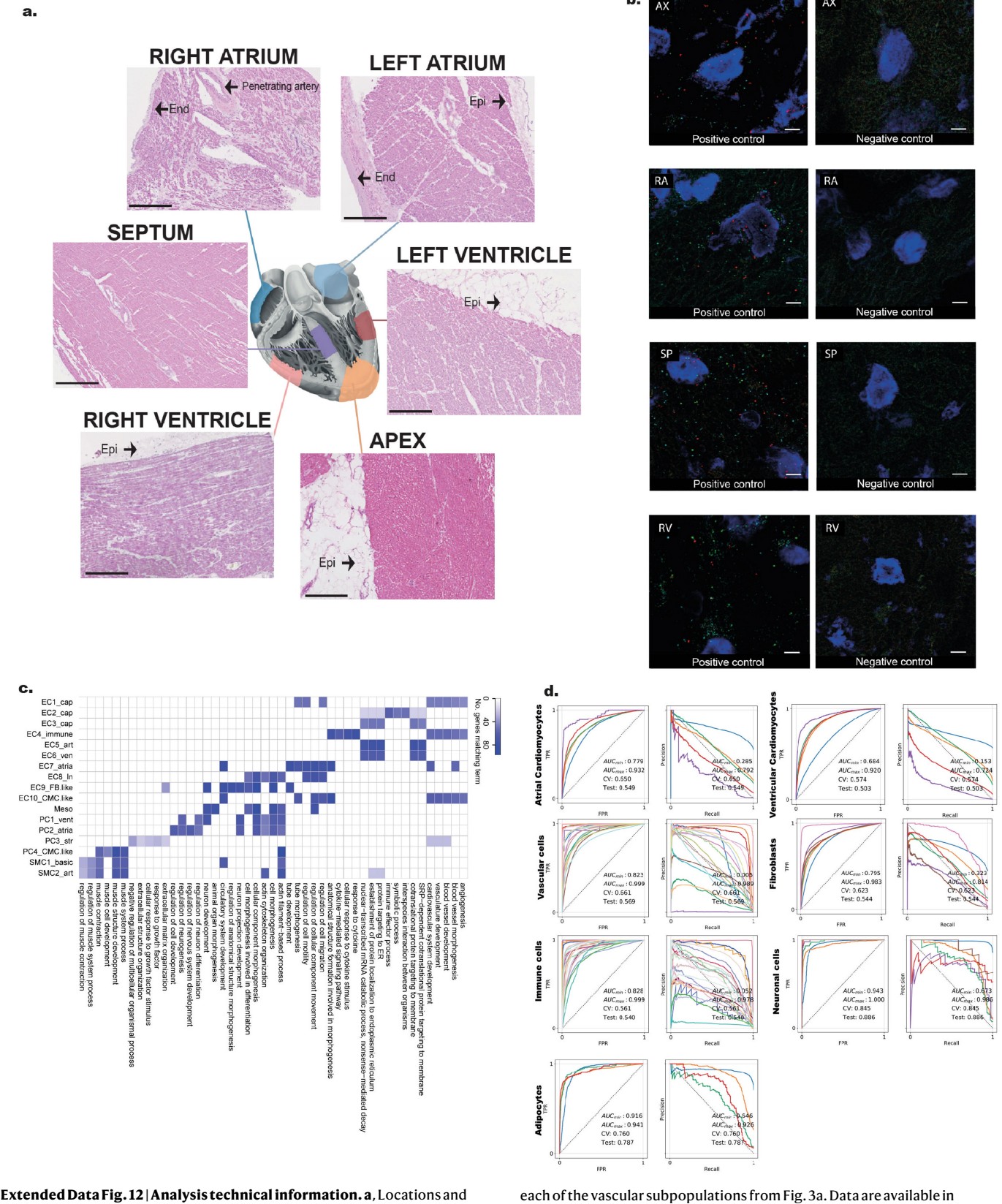

**Extended Data Fig. 12 | Analysis technical information. a**, Locations and representative histology section of six cardiac regions sampled, including right and left atrium, right and left ventricular free wall and left ventricular apex and interventricular septum. H&E, magnification ×10; scale bars, 500 μm. **b**, Spatial visualization of positive and negative RNAscope control probes. Scale bars, 5 μm. For statistics and reproducibility, see Methods. **c**, Heat map of top five significantly enriched Gene Ontology Biological Processes term for each of the vascular subpopulations from Fig. 3a. Data are available in Supplementary Table 25. **d**, SCCAF scores for each batch aligned manifold. For each population, we plotted the true positive (TPR) versus false positive (FPR) learning ratios from the subpopulation in each manifold. Next, we plotted how accurately the manifold represents each learned subpopulation based on the test training set and the CV cross-validation set. The closer the CV value to the test value, the better the manifold is at representing the subpopulations.

# Reporting Summary

Nature Research wishes to improve the reproducibility of the work that we publish. This form provides structure for consistency and transparency in reporting. For further information on Nature Research policies, see our Editorial Policies and the Editorial Policy Checklist.

## Statistics

For all statistical analyses, confirm that the following items are present in the figure legend, table legend, main text, or Methods section.

| n/a | Confirmed | |
|---|---|---|
| ☐ | ☒ | The exact sample size (*n*) for each experimental group/condition, given as a discrete number and unit of measurement |
| ☐ | ☒ | A statement on whether measurements were taken from distinct samples or whether the same sample was measured repeatedly |
| ☐ | ☒ | The statistical test(s) used AND whether they are one- or two-sided *Only common tests should be described solely by name; describe more complex techniques in the Methods section.* |
| ☐ | ☒ | A description of all covariates tested |
| ☐ | ☒ | A description of any assumptions or corrections, such as tests of normality and adjustment for multiple comparisons |
| ☐ | ☒ | A full description of the statistical parameters including central tendency (e.g. means) or other basic estimates (e.g. regression coefficient) AND variation (e.g. standard deviation) or associated estimates of uncertainty (e.g. confidence intervals) |
| ☐ | ☒ | For null hypothesis testing, the test statistic (e.g. *F*, *t*, *r*) with confidence intervals, effect sizes, degrees of freedom and *P* value noted *Give P values as exact values whenever suitable.* |
| ☒ | ☐ | For Bayesian analysis, information on the choice of priors and Markov chain Monte Carlo settings |
| ☒ | ☐ | For hierarchical and complex designs, identification of the appropriate level for tests and full reporting of outcomes |
| ☒ | ☐ | Estimates of effect sizes (e.g. Cohen's *d*, Pearson's *r*), indicating how they were calculated |

*Our web collection on statistics for biologists contains articles on many of the points above.*

## Software and code

Policy information about availability of computer code

| | |
|---|---|
| Data collection | Software used include: BD FACS Sortware 1.2.0.142 (BD Influx), Summit V5.4.0.16584 (BD XDP), BD FACSDIVA 6.1.3 (Aria), Axiovision 4.9.1 and ZEN 2.3 software (Zeiss), Harmony 4.9 (Perkin Elmer), Chromium Controller Firmware version 5.00 (10X Genomics). |
| Data analysis | Software used include: R 3.6, Python 3.7, 10X Genomics' Cell Ranger 3.0.2, Harmony 4.9 (Perkin Elmer), Axiovision 4.9.1 and ZEN 2.3 (Zeiss), Adobe Illustrator 24.2.3, CorelDraw X4, 10X Genomics' Space Ranger 1.0.0, ImageJ 1.52Q or 1.52P, Scanpy 1.4, bbknn 1.3.11, scGen 6c237d7, anndata 1.7, pandas 1.0.1, numpy 1.19, scVelo 1d87464. |

For manuscripts utilizing custom algorithms or software that are central to the research but not yet described in published literature, software must be made available to editors and reviewers. We strongly encourage code deposition in a community repository (e.g. GitHub). See the Nature Research guidelines for submitting code & software for further information.

## Data

Policy information about availability of data

All manuscripts must include a data availability statement. This statement should provide the following information, where applicable:
- Accession codes, unique identifiers, or web links for publicly available datasets
- A list of figures that have associated raw data
- A description of any restrictions on data availability

The data is made available through the Human Cell Atlas (HCA) Data Coordination Platform (DCP) and can be accessed here: https://www.ebi.ac.uk/ena/browser/view/ERP123138. The data can also be accessed and explored through the HCA Heart Project website at www.heartcellatlas.org.

# Field-specific reporting

Please select the one below that is the best fit for your research. If you are not sure, read the appropriate sections before making your selection.

☒ Life sciences ☐ Behavioural & social sciences ☐ Ecological, evolutionary & environmental sciences

For a reference copy of the document with all sections, see nature.com/documents/nr-reporting-summary-flat.pdf

# Life sciences study design

All studies must disclose on these points even when the disclosure is negative.

| | |
|---|---|
| Sample size | No sample-size calculation was done due to the nature of this study.<br>Non-failing hearts were collected from human donors from July 2018 to July 2019 on the basis of availability from CBTM (Cambridge, UK) and The University of Alberta (Canada). Our study explores the cellular composition of the healthy adult human heart and we state that the number of samples is not enough to make generalisations. |
| Data exclusions | No data were excluded from the analysis. For the final count matrix, we excluded cells based on pre-established criteria for single-cells: we excluded low quality samples and contaminating cells (i.e. - cells with low number of detected genes and high mitochondria content). |
| Replication | We performed single nuclei RNAseq on 14 hearts (4 - 6 regions each) and single cell RNAseq on 7 hearts (4 - 6 regions each), with comparable results among all the donors. The same samples were used for the validation experiments. The micrographs in Figure 2g, 3c/h and Extended Data Figure 2c (HAMP), 2e (CNN1), 3f, 3k, 4f are repeated with similar results in 2 individual tissue sections. The micrographs in Figure 2h, 3f and Extended Data Figure 2c (CNN1), 2e (PCDH7), 4e, 4h, 6d are repeated with similar results in 3 individual tissue sections. The micrographs in Figure 1e, 2d, 3e, and Extended Data Figure 1f, 2c (FHL1) and 4d are repeated with similar results in 4 individual tissue sections. The micrographs in Figure 2c and Extended Data Figure 2c (PRELID2), 7e, 9a are repeated with similar results in 6 or more individual tissue sections. Positive and negative controls were done once per used samples.<br><br>For skeletal muscle analysis, single cells and single nuclei were isolated from 5 individuals, with comparable results among all the donors. |
| Randomization | Only healthy individuals were considered in our analysis. Randomisation was not relevant due to the study design where non-failing hearts were used on availability. |
| Blinding | Only healthy individuals were considered in our analysis. Blinding was not relevant due to the study design where non-failing hearts were used on availability, and the analytical strategy would not benefit from it. |

# Reporting for specific materials, systems and methods

We require information from authors about some types of materials, experimental systems and methods used in many studies. Here, indicate whether each material, system or method listed is relevant to your study. If you are not sure if a list item applies to your research, read the appropriate section before selecting a response.

## Materials & experimental systems

| n/a | Involved in the study |
|---|---|
| ☐ | ☒ Antibodies |
| ☒ | ☐ Eukaryotic cell lines |
| ☒ | ☐ Palaeontology and archaeology |
| ☒ | ☐ Animals and other organisms |
| ☐ | ☒ Human research participants |
| ☒ | ☐ Clinical data |
| ☒ | ☐ Dual use research of concern |

## Methods

| n/a | Involved in the study |
|---|---|
| ☒ | ☐ ChIP-seq |
| ☐ | ☒ Flow cytometry |
| ☒ | ☐ MRI-based neuroimaging |

## Antibodies

| | |
|---|---|
| Antibodies used | anti-human CD45 monoclonal antibody-conjugated microbeads (Miltenyi Biotec, 130-045-801) in dilution 1:4 (20 ul of antibody-labeled microbeads in 80 ul of cell suspension buffer). |
| Validation | Commercially available product, full protocol and validation available at miltenyibiotec.com/_Resources/Persistent/25cf8ecca93dc183f1d96d5348e58ca0e9a07c40/DS130-045-801.pdf |

## Human research participants

Policy information about studies involving human research participants

| | |
|---|---|
| Population characteristics | Tissues were obtained form 14 individuals, eight (D1-7 and 11) collected in the United Kingdom and six (H2-7) collected in |

| | |
|---|---|
| Population characteristics | North America. The cohort consisted of seven male (D2, D3, D6, D7, H2, H3 and H4) and seven female (D1, D4, D5, D11, H5, H6 and H7) donors, in the range of 40-75 years of age. Six of the donors were classified as DCD (Donation after Circulatory Death, D2, D4-7 and D11) and eight donors were classified as DBD (Donation after Brain Death, D1, D3, H2-7). |
| Recruitment | Cardiovascular history was unremarkable for all donors, and this was the main recruitment criteria used for to include individuals in our study. We believe this method of recruitment does not represent any bias that can impact our results. |
| Ethics oversight | Heart tissues (D1-7 and 11) were obtained from deceased transplant organ donors after Research Ethics Committee approval (Ref 15/EE/0152, East of England - Cambridge South Research Ethics Committee) and informed consent from the donor families.<br><br>Heart tissues (H2-7) were obtained from deceased organ donors after Human Research Ethics Board approval Pro00011739 (University of Alberta, Edmonton, Canada). Informed consent from donor families was acquired via the institutional Human Organ Procurement and Exchange Program (HOPE). |

Note that full information on the approval of the study protocol must also be provided in the manuscript.

# Flow Cytometry

## Plots

Confirm that:

☒ The axis labels state the marker and fluorochrome used (e.g. CD4-FITC).

☒ The axis scales are clearly visible. Include numbers along axes only for bottom left plot of group (a 'group' is an analysis of identical markers).

☒ All plots are contour plots with outliers or pseudocolor plots.

☒ A numerical value for number of cells or percentage (with statistics) is provided.

## Methodology

| | |
|---|---|
| Sample preparation | As described in the Methods section. Briefly, the single nuclei were isolated by mechanical homogenisation and washed. The nuclei were stained with commercially available Hoechst 33342 dye (NucBlue , R376050). The samples were kept on ice and directly loaded onto the FACS-sorter. |
| Instrument | Becton Dickinson (BD) Influx, XDP, or FACSAria |
| Software | Proprietary software of the selected sorter. |
| Cell population abundance | N/A |
| Gating strategy | Single nuclei were selected for single signal on the SCC and FCC to avoid aggregates. The Hoechst-positive nuclei were selected without any size limit. The gating strategy is available as Supplementary Figure 1. |

☒ Tick this box to confirm that a figure exemplifying the gating strategy is provided in the Supplementary Information.

