## [Peer Review File · Nature]

Manuscript Title: Cells of the adult human heart

Editorial Notes:

Redactions – Mention of other journals

This document only contains reviewer comments, rebuttal and decision letters for versions considered at *Nature*. Mentions of the other journal have been redacted.

Reviewer Comments & Author Rebuttals

Reviewer Reports on the Initial Version:

Referee #1 (Remarks to the Author):

This well written manuscript describes work around single cell and single nuclei RNA sequencing of the normal adult human heart, analyzing 6 anatomic locations (atria, ventricles, septum, apex). This is one of many efforts applying recent single cell resolution technology to map organs in different species. The human heart is somewhat more challenging to investigate this way due to the size of cardiomyocytes and the difficulty of obtaining steady-state, healthy myocardium in sufficiently fresh quality. This manuscript is among the first studies to succeed in the human heart. There are prior publications on the human embryo heart (e.g. <https://doi.org/10.1016/j.celrep.2019.01.079>), a study published in a Nature subjournal on human adult normal and failing hearts (<https://doi.org/10.1038/s41556-019-0446-7>) and similar efforts deposited in bioarchives. There are a number of comparable human studies published on other organs and diseases. Hence, while on the technological forefront, there is no major methodological novelty.

7 male and 7 female hearts were studied, with samples from differing depths. Single cell and single nuclear sequencing was performed. Immune cells were studied after enriching for CD45+ cells by FACS. Cell types were identified and enumerated for different heart regions. Females had higher relative numbers of cardiomyocytes, which to me is the only biologically interesting, high impact novelty in this work. Cell subsets are described and regional and sex differences in gene expression, which are validated with smFISH and RNAscope. Comparative analysis was done for skeletal muscle. GWAS traits were localized to cell types.

Overall, I have no doubt that these data are a helpful reference for the community. However, the manuscript suffers from the typical flaws of the recent flood of scRNAseq studies: these manuscripts are descriptive, i.e. numerous "new subsets" are described without studies that indicate that they are functionally of any importance. A particular worry is that the number of identified cell subsets is governed by chosen resolution parameters during data processing. Then again, perhaps it does not matter how many cell subsets are described if there is no rigorous functional annotation in follow up studies. By itself, the impact of this study is limited to the fact that the deposited data will allow referencing of pathological findings.

It would be nice to see a table for relative abundance of major cells and subsets, as such percentages will be interesting for the reader.

"In the resulting data, myeloid cells represented around 5% of cells" -> be more specific regarding which cells (is cells here leukocytes, all cells, non-cardiomyocytes...)

The section on immune cells remains particularly superficial with little connection or references to the state of the art. One of the conclusions states that there are heart specific macrophages, which is true for macrophages in every organ, and really far from novel. Prior work that described heart macrophage subsets, with lineage tracing in mice and even in humans after heart transplantation, are much more detailed and advanced in ascribing functional relevance.

The manuscript is also lacking in describing certain limitations, discussing the still limited sample size potentially not reflecting the complete diversity contained within human hearts; nuclear sequencing related problems (e.g., nuclei RNA is only a small percentage of total mRNA and differ from cytoplasmic RNA, methods for identification of nuclear doublets are imperfect, with relevance to cardiomyocytes).

The authors should examine the concordance of whole cell versus nuclear transcriptomes for the cell classes available.

Referee #2 (Remarks to the Author):

Litviňuková and colleagues present a single-cell transcriptome atlas of the human adult heart. The study profiled around 500k single cells and nuclei across various locations and multiple donors. The resulting dataset can be considered as a first, although already comprehensive, draft of a heart atlas, a highly valuable resource for the community and an important contribution to the Human Cell Atlas project. The authors designed an elegant study that included different sampling strategies to cope with challenges of isolating rare or large cell types. Extensive validation with smFISH was performed, confirming many of the newly described subpopulations. The sampling and analysis strategies are well-suited and adequately tailored to the underlying tissue and cell types. The following comments could help to improve the robustness and clarity of the manuscript.

1. The authors circumvent the challenging preparation of CM or nerve cell by using single nuclei sequencing and subsequent computational integration. However, also snRNAseq introduces composition biases, e.g. against immune and endothelial cells and cannot be considered as ground-truth for tissue composition. This has at least to be mentioned/discussed in the main text. Although the data integration used state-of-the-art tools, could the authors include some plots showing the successful integration of sc/snRNAseq and immune-enriched datasets?

2. For the differential gene expression analysis, the author decided for a 3-fold threshold. This seems rather arbitrary and a p-value cut-off would have been more appropriate. In line, concluding that two cluster have similar expression programs for the lack of genes differentially expressed >3-fold seems too superficial.

3. Could some of the new subtypes be confirmed by smFISH to exclude that technical sampling artefacts, such as stress response (vCM4) or doublets, resulted in specific transcriptional signatures. For example, can the authors exclude that the Stroma13 population is not a transition state, but instead a population of doublets between EC and PC cells?

4. The authors chose the strategy to use single/multiple gene markers for the interpretation and annotation of the clusters. I agree, that this is favourable to GO/GSEA/etc enrichment analysis, especially when interpreting subtle differences. However, marker gene annotations are highly supervised and biased by the respective analyst. No doubt the authors are experts in the field, but can they confirm that genes from the lists of subtype marker point to consistent conclusions and that the presented interpretation is robust? The title mentions transcriptional programs, which are certainly used to cluster cells, but not in the interpretation. I suggest revising the title to make it

more general.

5. The authors include an elegant experimental design by enriching for CD45+ immune cells. They mentioned that resident immune cells are identified by calculating the enrichment in perfused vs. non-perfused organs. However, it is unclear how this was used in downstream analyses.

6. Many of the conclusions are based on the predicted interaction between cell types. However, it is not clear how robust such predictions are, especially to determine differential cell interaction networks between tissue, e.g. muscle vs heart. Could a difference in composition result in a different power to detect such interactions, e.g. for smaller/underrepresented cell types?

7. It would be great to have more meta-data on the sampling time of the donor hearts and an analysis if delays in the sampling of the fresh tissues affected the expression/composition of the samples.

Author Rebuttals to Initial Comments:

Summary of new experimental data and computational analyses:

In order to address the reviewers' comments as best we could during the Covid19 lockdown, we have now included the following additional new data:

- (i) We now provide additional smFISH stainings, bringing our validation of cell populations by (multiplex) smFISH to a total of 11 populations: the validated populations previously included 4 cardiomyocyte and 3 vascular populations, and these are now complemented by new stainings for 3 fibroblast and 1 neuronal cell population. In addition, we have now validated cell-cell interactions between specific fibroblast and macrophage subpopulations (Figure 3G, 3F and Extended Figure 4).
- (ii) We now include additional computational validation and integration of our data with a publicly available 10X Genomics Visium spatial transcriptomics tissue section from left ventricle myocardium, which validates specific spatial localisation of four cell populations, including the following two additional populations in addition to the 11 populations validated by smFISH: LYVE1+ macrophages and arterial smooth muscle cells (SMC2_art) (Supplementary Figure C6 and E5). This brings the total number of cell populations that we validate using spatial methods to 13 cell populations.
- (iii) To contribute to our understanding of the current Covid19 pandemic, we have included a short section summarising the ACE2, TMPRSS2, CTSB and CTSL expression in our cell types, and now also provide smFISH staining showing ACE2 mRNA expression in cardiac pericytes and fibroblasts (Extended Figure 7).

- (iv) With respect to data interpretation, our analyses of immune, neuronal and adipocyte cell lineages have advanced considerably in terms of the level of detail that we now provide (Figure 4, Extended Figure 4 and 5).

Point-by-point response to Reviewer 1

1. *“This well written manuscript describes work around single cell and single nuclei RNA sequencing of the normal adult human heart, analyzing 6 anatomic locations (atria, ventricles, septum, apex). This is one of many efforts applying recent single cell resolution technology to map organs in different species. The human heart is somewhat more challenging to investigate this way due to the size of cardiomyocytes and the difficulty of obtaining steady-state, healthy myocardium in sufficiently fresh quality. This manuscript is among the first studies to succeed in the human heart.”*

We are very grateful for these thoughtful comments.

“There are prior publications on the human embryo heart (e.g. <https://doi.org/10.1016/j.celrep.2019.01.079>), a study published in a Nature subjournal on human adult normal and failing hearts (<https://doi.org/10.1038/s41556-019-0446-7>) and similar efforts deposited in bioarchives. There are a number of comparable human studies published on other organs and diseases. Hence, while on the technological forefront, there is no major methodological novelty.”

While the reviewer points out how challenging it is to study the human heart, including cardiomyocyte size and limited availability of normal tissues, we want to emphasize other features that underscore the importance of fully understanding the molecular properties of human heart cells. First, the heart is unique amongst all organs in its perpetual activity throughout life, without any substantial regeneration of cells, hinting that there must be an incredible transcriptional complexity and diversity to maintain healthy cell function despite insults that occur. Second, the heart must rapidly adapt to various physiological challenges. Third, the hemodynamic pressures in the right and left ventricles differ by over 100mm Hg, and as such each chamber requires specific intrinsic properties and cellular responses. We therefore expect that different cardiac cells, and sub-types with divergent transcriptional profiles, allow the heart to meet these challenges. For the first time, we identify many of these cell sub-types and thereby demonstrate the molecular basis for enormous plasticity of the heart. While we and others have made a start, a full understanding of these mechanisms will require considerable future studies - there is still much more to discover. However, our efforts provide almost twice the already available data combined¹, allowing an unprecedented overview of the cellular heterogeneity of the adult human heart.

Regarding the comment that there is “no major methodological novelty”, we respectfully disagree and highlight the following features of our work:

- (i) The scale of our analyses are truly novel. We report the cellular composition of 14 human hearts across six anatomical regions, including approximately half a million human heart cells.
- (ii) We believe this is the first time a tissue from any organism has been analysed using both single nuclei (sn) and single cell (sc) RNA-Seq. To integrate these datasets, we applied cutting edge computational tools (scGen, a deep generative autoencoder²).
- (iii) To robustly define stable transcriptomic cell clusters we use iterative logistic regression models (SCCAF, just reported in our recent publication³) as a means of defining cell states (Supplementary Methods, page 7). Our SCCAF strategy provides a considerable methodological advance for the cell atlas community, and this work is the first time it has been applied to a newly generated dataset to our knowledge.
- (iv) We performed extensive validation of identified cell populations using single-molecule fluorescence in situ hybridization (smFISH). We confirmed specific cardiomyocyte subtypes (e.g., aCM2, aCM3, vCM2, vCM3), vascular (EC5_art, EC6_ven), mesothelial cells and fibroblast (FB3, FB4 and FB5) and neuronal (N5) sub-populations as well as cell-cell interactions between fibroblast and macrophage sub-populations.
- (v) We study for the first time correlations between cell populations and between male and female heart donors, uncovering remarkable differences in tissue architecture.
- (vi) We expand our dual cell and nuclei approach to human skeletal muscle (intercostal muscle) and contrast the skeletal muscle and cardiac data. Again this dual profiling of two related tissues at both single cell and nuclei level is novel to our knowledge.

In summary, by employing novel methodologies, we provide fundamental new insights into the molecular basis for cardiac physiology.

2. *“7 male and 7 female hearts were studied, with samples from differing depths. Single cell and single nuclear sequencing was performed. Immune cells were studied after enriching for CD45+ cells by FACS. Cell types were identified and enumerated for different heart regions. Females had higher relative numbers of cardiomyocytes, which to me is the only biologically interesting, high impact novelty in this work. Cell subsets are described and regional and sex*

differences in gene expression, which are validated with smFISH and RNAscope. Comparative analysis was done for skeletal muscle. GWAS traits were localized to cell types.”

Thank you for recognizing the comprehensive approaches that are employed in this study. Of note, the enrichment of CD45+ cells was performed using magnetic separation (not FACS), as described in the Supplementary Methods section, page 3:

“CD45+ cell enrichment: Cell suspension is prepared as described above and subsequently are labelled using anti-human CD45 monoclonal antibody-conjugated microbeads according to the manufacturer’s protocol (Miltenyi Biotec). Briefly, up to 107 cells are incubated for 15 min at 4°C in 80 µl of PBS/BSA/EDTA buffer (1x PBS pH 7.2, 0.5% BSA, 2 mM EDTA) containing 20 µl CD45 microbeads. Cell suspension is washed in PBS/BSA/EDTA buffer once and harvested by centrifugation (330×g, 10 min, 4°C). Resuspended cells are applied to MACS LS columns (Miltenyi Biotec). CD45-depleted cell fraction is discarded after three washes with PBS/BSA/EDTA buffer and CD45+ cell fraction is collected in PBS/BSA/EDTA buffer by removal of the columns from the magnetic field. CD45+ cells are counted and resuspended in PBS/BSA/EDTA buffer to a concentration of $\geq 2 \times 10^6$ /ml before further processing using a Chromium Controller (10X Genomics) according to the manufacturer's protocol.”

3. *“Overall, I have no doubt that these data are a helpful reference for the community. However, the manuscript suffers from the typical flaws of the recent flood of scRNAseq studies: these manuscripts are descriptive, i.e. numerous “new subsets” are described without studies that indicate that they are functionally of any importance. A particular worry is that the number of identified cell subsets is governed by chosen resolution parameters during data processing. Then again, perhaps it does not matter how many cell subsets are described if there is no rigorous functional annotation in follow up studies. By itself, the impact of this study is limited to the fact that the deposited data will allow referencing of pathological findings.”*

The “numerous new subsets” reflects the definition of cell state and their fluidity. We defined cell states based on the combination of unbiased algorithms and manual curation using biological expertise within our group. However, we want to highlight the novel method SCCAF, reported in our recent publication³, that we employed to assess the statistical and computational robustness of transcriptomic cell clusters as a means of defining cell states. We used this approach to define the clusters in our dataset (Supplementary Methods) and visualised the results in Supplementary Figure H1:

Supplementary Figure H1: SCCAF scores for each batch aligned manifold. For each population, we plot the True Positive (TPR) versus False Positive (FPR) learning ratios from the subpopulation in each manifold. Next we plot how accurately the manifold represents each learned subpopulation based on the Test training set and the CV cross-validation set as calculated from the confusion matrix. The closer the CV value to the Test value, and the closer both are to 1, the better the manifold is at representing the subpopulations, as explained in Miao et al (2020)³. A. Atrial Cardiomyocytes, B. Ventricular Cardiomyocytes, C. Vascular cells, D. Fibroblasts, E. Immune, F. Neuronal, G. Adipocytes.

While we agree that understanding specific functions of particular subpopulations is an important topic for future investigation, this will undoubtedly take considerable time. With regard to cardiomyocytes, we hope that the reviewer recognizes that there are no immortalized cardiomyocyte lines, nor methods to identify and separate subpopulations in living heart tissues, so creative strategies for studying our newly defined cardiomyocyte cell types mechanistically in vitro will need to be developed going forward.

However, we respectfully disagree that “it does not matter how many cell subsets are described if there is no rigorous functional annotation in follow up studies.” First, the observation of higher cardiomyocyte numbers in female hearts indicate gender-specific differences in tissue architecture that are likely to be biologically meaningful given the gender differences in susceptibility to heart disease.

Second, we now provide extensive smFISH validation of our findings beyond those performed at the time of first submission (Figure 3E- F, Extended Figure 4). In addition, we now provide initial integration of our data with comparison to publically available 10x Genomics Visium data (Supplementary Figure C6 and E5). These techniques validate the spatial localisation of cell populations. The distinct transcriptional profiles in cells with variable percentages in particular chambers of the heart, combined with the very different hemodynamics within chambers certainly provides an initial functional annotation and interpretation. Similarly, finding that in vivo ECM-producing fibroblasts in atrium and ventricle differ indicates that fibrotic processes are regionally distinct, and very likely contribute to the very high prevalence of atrial arrhythmias in the general population. In summary, our data not only defines the number and characteristics of cell subtypes but also provides functional attributions that illuminate cardiac physiology and will help to decipher cardiac disease.

4. *“It would be nice to see a table for relative abundance of major cells and subsets, as such percentages will be interesting for the reader.”*

In the original version (and maintained in this revised version), the distribution of the major cell types per source is included in Supplementary Table A6 shown here:

Percentage / source	Nuclei	Cells	CD45
vCardiomyocytes	38.0	0.0	0.0
aCardiomyocytes	7.1	0.0	0.0
Fibroblasts	17.1	4.9	0.8
Endothelial	8.6	69.2	53.1
Pericytes	18.1	13.7	15.4
Smooth Muscle	2.9	6.6	4.9
Lymphoid	2.1	2.3	12.7
Myeloid	3.7	2.5	12.7
Neuronal	1.0	0.5	0.4
Adipocytes	1.2	0.0	0.0
Mesothelial	0.2	0.3	0.0

Supplementary Table A6: Distribution of major cell types across the three sources of data, highlighting the different profiles captured for cells and nuclei.

and these distributions were previously and are still visualised as bar plots per source in Extended Figure 1C, as shown below:

Extended Figure 1: Source contribution to the major cell types C. Barplot representing the distribution of cell types obtained by each source. The raw data are available in the Supplementary Table A6.

In addition, the details for the subpopulations belonging to each major cell lineage can be found in Supplementary Tables B-E.

5. *“In the resulting data, myeloid cells represented around 5% of cells” -> be more specific regarding which cells (is cells here leukocytes, all cells, non-cardiomyocytes...)”*

We apologize for this lack of specificity in this section. In order to provide a more detailed description of the cardiac immune population we have now removed that sentence, for brevity, and provided a more detailed description of the proportion of cell states between immune cells and between cardiac compartments. The reviewer can now see that myeloid populations account for 4.73 % of the total cardiac cells captured by the four different data sources in our study. The complete information is now available as Supplementary Table E1:

Cell States	Harvard-Nuclei	Sanger- CD45	Sanger- Cells	Sanger-Nuclei	Total cell state	% of immune cells	% of all cells
NK	442	2879	237	70	3628	8.88	0.75
CD16+Mo	607	2320	253	98	3278	8.02	0.67
DOCK4+MΦ1	1178	31	0	2030	3239	7.93	0.67
CD4+T_cytox	816	1608	162	527	3113	7.62	0.64
LYVE1+MΦ1	628	1161	275	954	3018	7.38	0.62
CD8+T_tem	1459	838	81	623	3001	7.34	0.62
CD8+T_cytox	1029	1565	288	74	2956	7.23	0.61
LYVE1+MΦ2	241	553	40	1254	2088	5.11	0.43
LYVE1+MΦ3	414	777	59	707	1957	4.79	0.40
Mo_classic	163	1524	163	33	1883	4.61	0.39
Mo_pi	207	1214	47	184	1652	4.04	0.34
DOCK4+MΦ2	757	4	0	865	1626	3.98	0.33
Mast	1222	97	13	211	1543	3.78	0.32
NKT	257	1034	70	102	1463	3.58	0.30
MΦ_mod	56	1077	83	97	1313	3.21	0.27
MΦ_AgP	81	996	103	98	1278	3.13	0.26
B_cells	380	642	71	102	1195	2.92	0.25
CD4+T_naive	109	762	114	62	1047	2.56	0.22
dpT	251	317	24	222	814	1.99	0.17
doublers	205	152	12	254	623	1.52	0.13
NØ	2	36	83	0	121	0.30	0.02
IL17RA+Mo	32	0	0	0	32	0.08	0.01

Supplementary Table E1: Distribution of immune cells across the four data sources and their proportions considering all immune cells (% of immune cells) and considering all cells(% of all cells) across all six cardiac tissue locations.

6. *“The section on immune cells remains particularly superficial with little connection or references to the state of the art. One of the conclusions states that there are heart specific macrophages, which is true for macrophages in every organ, and really far from novel. Prior work that described heart macrophage subsets, with lineage tracing in mice and even in humans after heart transplantation, are much more detailed and advanced in ascribing functional relevance.”*

We are grateful to the reviewer for these comments, although we are not clear to which works he/she is referring. In our revised manuscript we provide a refined annotation of the immune cells (Figure 4, Supplementary Table E1), and further analysed the transcriptional profile of immune cells involved in cardiac homeostasis. We have re-classified the Foam cells to MΦ_AgP, which is a more specific description given their enrichment for HLA class II genes. We do observe the expression of TREM2, which has been described as part of the signature of lipid-associated MΦ (LAM)⁴.

A new section compares the transcriptional profiles of cardiac immune cells with previous studies on cardiac-resident MΦ⁵ and development⁶, suggesting that these populations might be yolk sac-

derived, though this requires further computational and experimental investigation (Supplementary Figure E2).

The MΦ compartment includes three LYVE1+MΦ populations (LYVE1+MΦ1-3), two of which enrich for clathrins and cathepsin genes (LYVE1+MΦ1-2) while LYVE1+MΦ3 enriches for HLA-DOA, HLA-DQA1-2, HLA-DQB1. LYVE1+MΦ appear related to recently described murine tissue-resident populations associated with cardiac^{5,7} and aortic remodeling⁸. However, unlike their murine counterparts, none of these human cardiac MΦ populations express TIMD4 (Supplementary Figure E2b)⁷. Monocyte-derived MΦs (MΦ_mod) cluster separately from LYVE1+MΦ, and although they also express LYVE1 and FOLR2, in addition they have the monocyte-like markers CEBPB and S100A8, as well as the chemoattractant cytokines CCL13 and CCL18. Antigen-presenting MΦ (MΦ_AgP) do not express FOLR2, LYVE1 or MERTK, but express HLA-DRA, HLA-DMA, HLA-DMB and HLA-DPA1 genes and TREM2, previously described in lipid-associated MΦ (LAM)⁴ (Supplementary Figure E2b). We also detect two populations of DOCK4+MΦ that do not express Q1CA or FOLR2, which are distinct from each other via higher expression of IL4R, STAT3 and ITGAM in DOCK4+MΦ1 compared with DOCK4+MΦ2 (Supplementary Figure E2c).

Supplementary Figure E2b: Expression (log₂FC) of LYVE1, FOLR2 and TIMD4 characteristic of the self-renewing tissue-resident murine macrophages described in Dick SA et al 2019, as well as MERTK as described in Bajpai G et al 2018 and the TIMD2 expression associated to Lipid-Associated Φ (LAM) described in Jaitin D et al 2019. Complete signatures can be found in Supplementary Table E7.

Supplementary figure E2c: Scaled expression (log2FC) of genes differentiating DOCK4+MΦ1 from DOCK4+MΦ2: IL4R, ITGAM, STAT3, DOCK1, HIF1A, RASA2.

We also infer cell-cell interactions for these macrophage populations from single cell transcriptomes, as shown in the figure below (Figure 4B and Supplementary Figure E3).

Supplementary Figure E3: Cell-Cell interactions enriched for “Extracellular matrix organisation” for cardiac fibroblasts, myeloid cells and cardiomyocytes . Mean: Mean of combined gene expression of interacting pairs (log2FC).

Finally, using an external spatial transcriptomics 10X Genomics Visium dataset from left ventricle myocardium, we have now mapped the co-expression of ligand-receptor pairs for FB4-M Φ interactions and their tissue localisation in the human heart, as shown below (Supplementary Figure E5).

Supplementary figure E5: Spatial mapping of the CD74_MIF interaction between LYVE1+M Φ and FB4 on a publicly available 10X Genomics Visium dataset for LV myocardium. We identified four spots where we observe co-expression of FN1, LYVE1, CD74 and MIF, as predicted from the cell-cell interactions. The bar represents log2FC.

We agree with the reviewer's comment that tissue-resident myeloid cells are present across all human organs. Importantly, our analyses reveal the richness and complexity of human cardiac-resident macrophages for the first time. . This reference data on cardiac-specific immune cells provides a starting point for future work dissecting mechanisms of cardiac homeostasis in more detail.

7. *“The manuscript is also lacking in describing certain limitations, discussing the still limited sample size potentially not reflecting the complete diversity contained within human hearts; nuclear sequencing related problems (e.g., nuclei RNA is only a small percentage of total mRNA and differ from cytoplasmic RNA, methods for identification of nuclear doublets are imperfect, with relevance to cardiomyocytes).”*

We agree that further studies of healthy hearts are needed, particularly to determine the influences of age, gender, ancestries and other parameters that vary considerably across humans. We indicate this in the Discussion page 20 as follows:

“We are aware of the limitations of the cell capture by different data sources and that our results may be biased based on our surgical sampling, but we expect that our results will inform studies of other cardiac regions (valves, papillary muscle, conduction system), propel studies with large cohorts to elucidate the roles of age, gender, and ancestry on normal cardiac physiology and provide critical insights to enable mechanistic understanding of heart disease. “

Nevertheless, we also recognize the following major strengths of our study:

- (i) This is the largest human cardiac single cell study to date, capturing the cellular diversity of the human heart by combining single cells and single nuclei data with ~500K cells, which allows us to explore transcriptional differences between healthy cardiac cell states with unprecedented resolution.
- (ii) The combination of single cells and nuclei provide a more comprehensive overview of the cell populations in the heart than any other study to date. Our findings are based on these two complementary methodologies that differentially capture cardiac cell types, and provide a robust harmonised overview of heart cell populations. Additionally, using this methodology we are able to confirm our findings across two geographically distinct sites (UK and North America).

Regarding the technical limitations of snRNA-Seq, we are delighted that we have orthogonal technologies that validate our findings, including the scRNA-Seq data, the smFISH data and the 10x Genomics Visium spatial transcriptomics data.

8. *“The authors should examine the concordance of whole cell versus nuclear transcriptomes for the cell classes available.”*

This question relates to the methodological novelty of analysing the same tissue sample using both scRNAseq and snRNAsesq, which we articulate in points 1 and 7 above.

Overall, by using variational autoencoders (scGen2), we make sure that the transcriptional features that are present only in cells or nuclei are properly integrated in latent space. This provides an

unbiased platform to interrogate the data while accounting for differences in the data sources. This reveals global concordance between the data types (cell and nuclei), in the sense that they integrate well using deep learning methods.

We do indeed see a difference in the composition of cell types captured by the different methods, and this is now summarised in Extended Figure 1 and Supplementary Figure A4.

Extended Figure 1: Source contribution to the major cell types A. UMAP representation of the major cell types colored by source. B. Individual UMAP representations highlighting each of the sources (UK = United Kingdom, NA = North America). C. Barplot representing the distribution of cell types obtained by each source.

For this precise reason we believe that the usage of cells and nuclei provide a complementary approach. We have now added the following statement to the Supplementary Methods section, page 8:

“Distributions of dispersed cells and isolated nuclei: The different procedures for obtaining isolated nuclei and dispersed cells result in significantly different distributions of cell types (Supplementary Table A2, A5 and Supplementary Figure A4). Notably, 30.1% and 49.2% of

isolated nuclei are derived from aCM and vCM, while these cells were mostly excluded from preparations of isolated and CD45 selected cells.

Excluding CM, the distribution of cell types identified from isolated nuclei and dispersed cells remain distinct (Supplementary Table A5). While 59.0% of dispersed cells are EC, only 15.7% of nuclei are derived from EC. By contrast, 64.2% of nuclei are from FB (31.2%) and PC (33.0%) while only 17.1% of dispersed cells are FB (2.3%) and PC (14.8%). These differences may reflect sensitivity of EC nuclei to isolation procedures or resistance of PC and FB to cellular enzymatic digestion.

Despite differences in cell distributions between isolated nuclei and dispersed cells the gene expression profiles of cell lineages are reasonably correlated ($r > 0.4$ for each cell type). To address the concordance of the genes captured by cells and nuclei, we compared the expression of the major cell type markers from Figure 1C across the 4 sources (Supplementary Figure A4). As nuclei lack cytoplasmic RNA, the expression of certain genes, especially immune genes NK7G and C1QA, was lower in nuclei compared to the cells. Nevertheless, the general trend with respect to marker genes was consistent across the two sources, and the same genes distinguished individual cell types independent of the source.”

Point-by-point response to Reviewer 2

1. *“The authors circumvent the challenging preparation of CM or nerve cell by using single nuclei sequencing and subsequent computational integration. However, also snRNAseq introduces composition biases, e.g. against immune and endothelial cells and cannot be considered as ground-truth for tissue composition. This has at least to be mentioned/discussed in the main text.”*

As the reviewer points out correctly, both sc and snRNA-seq introduce capture bias, for example when it comes to the capturing of endothelial cells and immune cells. We now comment on this in the Introduction, page 3:

“We incorporate snRNA-Seq to ensure high throughput capture of large cardiomyocytes (length/width: ~100/25 μ m) and scRNA-Seq to upsample and enrich endothelial and immune cell populations.”

And we address the limitation in the Discussion section (highlighted in response to Reviewer 1-7 above), and in the Supplementary Methods section (highlighted in response to Reviewer 1-8 above).

2. *“Although the data integration used state-of-the-art tools, could the authors include some plots showing the successful integration of sc/snRNAseq and immune-enriched datasets?”*

For global integration, we refer to the response directly above, and response to Reviewer 1-8 above. For the answer to this question relating specifically to immune cells, we provide the analysis below for the reviewer.

Reviewer Figure 1: Integration of data sources for the immune cells using scGen. A. Manifold of 40,868 myeloid and lymphoid cardiac cells. NØ, neutrophils; NK, Natural killer cells; NKT, Natural killer T cells; CD4+T_naive, Naive CD4+ T cells; CD4+T_ctl, CD4+ cytolytic T cells; CD8+T_tem, CD8+ effector-memory T cells; CD8+T_cytox, CD8+ cytotoxic T cells; dpT: CD4+CD8+ T cells; Mo_classic, Classical monocytes; CD16+Mo, CD16+ monocytes; Mo_pi, pro-inflammatory monocytes; IL17RA+Mo, IL17RA+ monocytes; MØ_AgP, HLA class II antigen-presenting MØ; MØ_mod, monocyte-derived MØ; LYVE1+MØ1-3, M2-like, LYVE1+ MØ set 1 - 3; DOCK4+MNP_1, DOCK4+ Mononuclear phagocyte 1; DOCK4+MNP_2, DOCK4+ Mononuclear phagocyte 2; doublets: cell doublets. B. Cell states that are captured preferentially by a specific data source (e.g.: Mast cells mostly in nuclei from both Harvard and Sanger) are harmonised

with scGen. Harvard-Nuclei, Single nuclei samples prepared at Harvard Medical School; Sanger-Nuclei, Single nuclei samples prepared at Wellcome Sanger Institute; Sanger-Cells, Total single cell samples prepared at Wellcome Sanger Institute; Sanger-CD45, CD45+enriched samples prepared at Wellcome Sanger Institute. C. Our approach allows for the interrogation of cell states from all four different data sources, regardless of their per-source proportion, providing a unified picture of cardiac immune cells.

3. *“For the differential gene expression analysis, the author decided for a 3-fold threshold. This seems rather arbitrary and a p-value cut-off would have been more appropriate. In line, concluding that two cluster have similar expression programs for the lack of genes differentially expressed >3-fold seems too superficial.”*

There is a misunderstanding here: we did not use a 3-fold threshold for our differentially expressed gene analysis. Instead, we used a p-value cut-off to define differential gene expression and specified this in the in the Supplementary Methods, page 9:

“Detection of differentially expressed genes (DEGs): To help with the annotation of the subpopulations of each cell compartment, we calculated the DEGs using the Wilcoxon Rank Sum test as implemented in the scanpy workflow and recommended by recent benchmarking studies 9. A gene was considered to be differentially expressed if it has a $\log_2FC > 1$ and a p-value $< 1e-05$, unless stated otherwise in the analysis section.”

In the first submission, on page 7, we wrote: "Of 10,228 genes expressed among all CM, 389 genes showed >3-fold higher expression in aCM..." This was just to provide an example to illustrate what magnitude in fold changes is captured in the data. We have now removed this wording to avoid misunderstandings.

4. *“Could some of the new subtypes be confirmed by smFISH to exclude that technical sampling artefacts, such as stress response (vCM4) or doublets, resulted in specific transcriptional signatures. For example, can the authors exclude that the Stromal3 population is not a transition state, but instead a population of doublets between EC and PC cells?”*

We agree with the reviewer that confirmation of cell subtypes identified by sc/sn RNA-Seq is important. As suggested and described in response to Reviewer 1-3 above, we have now further extended our validations using smFISH for confirmation of the fibroblast populations and other populations.

For the vCM4 cluster our analysis shows no indication of this being due to stress response: no classical stress response genes are activated. Neither is there evidence of doublets: no increase in scrublet score, etc. Due to limited access during the COVID19-lockdown we were not able to proceed with smFISH for validation of this cluster.

The Stromal3 (now PC3_str) population also shows no increased scrublet score. While n_gene counts and n_reads counts in the cells of this cluster are below the stringent thresholds used throughout

the study and in published reports, the values are amongst the highest of clusters in our study. Although we cannot fully exclude the possibility that doublets of EC and PC cells contribute to this cluster, the size of the cluster and the fact that both nuclei and cells contribute to this cluster suggests that it represents a real cell state. We have amended the manuscript text accordingly in Supplementary Methods, page 9 :

“The PC_str contains similar contribution of cells and nuclei, that has a scrublet score below the stringent threshold used, nevertheless the average number of genes and counts in the cell fractions is higher than average, thus, despite our stringent quality filtering, we cannot exclude there might be doublets in this cluster.”

5. *“The authors chose the strategy to use single/multiple gene markers for the interpretation and annotation of the clusters. I agree, that this is favourable to GO/GSEA/etc enrichment analysis, especially when interpreting subtle differences. However, marker gene annotations are highly supervised and biased by the respective analyst. No doubt the authors are experts in the field, but can they confirm that genes from the lists of subtype marker point to consistent conclusions and that the presented interpretation is robust? The title mentions transcriptional programs, which are certainly used to cluster cells, but not in the interpretation. I suggest revising the title to make it more general.”*

We agree with the reviewer - de novo interpretation of cell subtypes is indeed an intellectually challenging task. To avoid bias, we annotated clusters not only based on individual genes, but took multiple genes and gene expression programmes into account, as well as considering information from the cellular context including cell-cell interactions. We explain this in detail in the Supplementary Methods section, page 8:

Detection of differentially expressed genes (DEGs): To help with the annotation of the subpopulations of each cell compartment, we calculated the DEGs using the Wilcoxon Rank Sum test as implemented in the scanpy workflow and recommended by recent benchmarking studies 9. A gene was considered to be differentially expressed if it has a $\log_2FC > 1$ and a p-value $< 1e-05$, unless stated otherwise in the analysis section.

Cell-Cell interactions: Expression matrices of the populations considered were exported from the AnnData, together with a metadata table that contains the cell-barcodes as indices. We then run CellPhoneDB as follows: `cellphonedb method statistical_analysis meta.tsv counts.tsv --subsampling --subsampling-log True--subsampling-num-cells 50000 --counts-data=gene_name --threads = 60`. CellPhoneDB raw predictions were filtered by removing those interactions with a p-value $> 1.0e-05$. Significant pairs were then submitted for gene set enrichment analysis into ReactomeDB, enrichR and ToppFun for functional classification. The vascular cells were randomly sub-sampled to 39 000 cells before the analysis.

As recommended, we have now revised the title to “Cells of the adult human heart”.

6. “The authors include an elegant experimental design by enriching for CD45+ immune cells. They mentioned that resident immune cells are identified by calculating the enrichment in perfused vs. non-perfused organs. However, it is unclear how this was used in downstream analyses. “

We appreciate the reviewer's comment on the need to clarify the analysis approach. In order to define tissue-resident immune cell populations, we compare the transcriptional signature of tissue-resident immune cells from previous studies 5–7 to the populations in our data. We now provide these references and the markers in Supplementary Table E6:

Study	Feature	Genes
Bajpai G et al 2018	Tissue residence	CCR2, MERTK, CD68, CD64, CD14, PTPRC, FCGR1A, FCGR1B, CD33, CD163, HLA-DRA
Dick SA et al 2019	Tissue remodelling	FOLR2, LYVE1, TIMD4, C1QA, CD14
Bian Z et al 2020	Yolk sac-derived	CD24, DHCR24, ARPC1A, LDHA, PGAM1, ATP5B, EIF5A, KPNA2, PLVAP, PRTG, RHOA, CALM1, CD63, S100A10, BZW1, S100A4, KRT18, NDUFV3, DSC2, P4HB, CISD1, MYDGF, F2R, GAPDH

Supplementary Table E6: Transcriptional signatures for cardiac tissue resident M ϕ from Bajpai G *et al* 2018⁵, Self renewing tissue macrophages from Dick SA *et al* 2019⁷ and Yolk sac-derived macrophages from Bian Z *et al* 2020⁶.

Using this information, we prioritise populations with support for tissue residency from the data, and hypothesise that they may be involved in cardiac homeostasis. (Note that we do not distinguish between different classes of transplant donors; apologies for the misunderstanding regarding this point.) Using these signatures we found five populations of M ϕ (LYVE1+M ϕ 1-3, M ϕ _mod and M ϕ _AgP) with expression of markers associated with tissue residency and tissue repair (LYVE1, FOLR2).

This information about the methods used has now been clarified and expanded in the Results section and supplementary information, including the figure panel a from Supplementary Figure E2 shown below (and including Supplementary Table E1 and Supplementary Methods).

Supplementary Figure E2a: Visualisation of transcriptional signatures from published studies. The values in the key represent the likelihood of the external transcriptional signature to be present when comparing it against the transcriptional background of a cardiac immune population. Bajpai_2018 = CCR2-MERTK+ Tissue resident macrophages from Bajpai G et al 2018., Dick_2019 = Self renewing tissue macrophages from Dick SA et al 2019, Bian_2020 = Yolk sac-derived macrophages from Bian Z et al 2020. The complete signature can be found in Supplementary Table E6.

7. *“Many of the conclusions are based on the predicted interaction between cell types. However, it is not clear how robust such predictions are, especially to determine differential cell interaction networks between tissue, e.g. muscle vs heart. Could a difference in composition result in a different power to detect such interactions, e.g. for smaller/underrepresented cell types?”*

It is correct that small/underrepresented cell types will provide less statistical power for inference of interactions. Given the large datasets in this work, we have sufficient statistical power for discovery of several novel interactions, some of which we are able to validate with spatial data. This now includes the following new validations of interactions since submission of the manuscript:

- EC5_art and SMC2_art interaction via JAG1_NOTCH2 validated using Visium as shown in Supplementary Figure C6 illustrated below
- FB3 and MΦ interaction via OSMR_OSM validated using smFISH as show in Figure 3F below
- FB4 and LYVE1+MΦ3 interaction via MIF_CD74 validated using Visium as shown in Supplementary Figure E5 below

Supplementary Figure C6: Marker genes and interacting receptors and ligands visualized on the 10x Genomics Visium section. A. Visualization of expression (\log_2FC) of CDH5 (pan-EC marker), SEMA3G and GJA5 (EC5_art markers). The zoom on the middle area shows an arteriole passing through the tissue. B. Visualization of expression (\log_2FC) of ACKR1 and PLVAP (EC6_ven markers). The zoom on the corner area shows a venule potentially passing

through the tissue. C. Visualization of expression (log2FC) of MYH11 and ACTA2 (pan-SMC markers). The zoom on the middle area, corresponding to the middle area in A., shows co-occurrence of SMC and EC5_art. D. Visualization of JAG1 and NOTCH2 expression (log2FC), the predicted interaction partners for arterial EC and SMC. The zoom of the middle area - corresponding to A. and C. - shows co-occurrence of those genes, supporting the cell-cell interactions prediction.

Figure 3: Vascular,stromal and mesothelial cells. F. Close-proximity of macrophages (MΦ) and OSM-pathway active FB3 suggests cross-talk between both cell types. The list of OSM-pathway genes can be found in Supplementary Table D2.

Supplementary figure E5: Spatial mapping of the CD74_MIF interaction between LYVE1+MΦ and FB4 on a publicly available 10X Genomics Visium dataset for LV myocardium. We identified four spots where we observe co-expression of FN1, LYVE1, CD74 and MIF, as predicted from the cell-cell interactions. The bar represents log2FC.

8. *“It would be great to have more meta-data on the sampling time of the donor hearts and an analysis if delays in the sampling of the fresh tissues affected the expression/composition of the samples.”*

We thank the reviewer for this important comment and agree that the sampling method and sampling time is crucial to capture clean and unbiased data. We reviewed the meta-data to address this concern. Samples collected in North America were snap-frozen within minutes of collection, while samples collected in the UK were placed in cold University of Wisconsin Solution in ice within minutes, before transfer to cardioplegic solution and delivery to the research lab for fresh cell isolation and snap freezing (details are now included in the Supplementary Methods). Although, differences in the time from sampling to freezing/tissue dissociation could occur in theory, data from two geographically distinct regions and independent labs have generated largely reproducible data. Previous studies have shown minor changes in tissues within the first 24 hours when stored in cold conditions and we have now added further detail in the Supplementary Methods section as follows (page 2):

“All our tissues were stored and transported on ice at all times until freezing or tissue dissociation to minimise any transcriptional degradation. Previous studies on the post-mortem tissue stability of the GTEx consortium on bulk tissues 10 and in single cells 11 suggest only minor changes in tissues within the first 24h post-mortem when stored in cold conditions.”

In addition, for clarity, we have expanded on the precise protocols for deceased transplant donation and tissue acquisition that were used in this work in the Supplementary Methods section (page 2) as follows:

“Tissue is acquired from UK and North American donors (D1-7 and 11, H2-7) after circulatory death (DCD: D2, D4-7 and D11) and after brain death (DBD: D1, D3, H2-7). For UK DCD donors after a five minute stand-off and for DBD, the chest is opened, the aorta is cross-clamped and cardiac samples are acquired. For North American DBD donors, the aorta is cross-clamped, cold cardioplegia (Celsior) is administered under pressure via the aorta to arrest beating, the heart is excised, rinsed in cold saline and samples acquired. All donor samples are full-thickness myocardial biopsies from LA, RA, RV, SP, AX and LVs, with intentional exclusion of large epicardial fat deposits. Samples used for single nuclei isolation are flash-frozen and stored at -80°C. Single cell isolation and CD45+ enrichment is carried out on freshly collected samples. Residual tissue after nuclei and cell isolation procedures is formalin-fixed or frozen in OCT for additional studies.”

References:

1. Tucker, N. R. et al. Transcriptional and Cellular Diversity of the Human Heart. *bioRxiv* 2020.01.06.896076 (2020) doi:10.1101/2020.01.06.896076.
2. Lotfollahi, M., Wolf, F. A. & Theis, F. J. scGen predicts single-cell perturbation responses. *Nat. Methods* 16, 715–721 (2019).
3. Miao, Z. et al. Putative cell type discovery from single-cell gene expression data. *arXiv [q-bio.QM]* (2020).
4. Jaitin, D. A. et al. Lipid-Associated Macrophages Control Metabolic Homeostasis in a Trem2-Dependent Manner. *Cell* 178, 686–698.e14 (2019).
5. Bajpai, G. et al. The human heart contains distinct macrophage subsets with divergent origins and functions. *Nat. Med.* 24, 1234–1245 (2018).
6. Bian, Z. et al. Deciphering human macrophage development at single-cell resolution. *Nature* (2020) doi:10.1038/s41586-020-2316-7.
7. Dick, S. A. et al. Self-renewing resident cardiac macrophages limit adverse remodeling following myocardial infarction. *Nat. Immunol.* 20, 29–39 (2019).
8. Lim, H. Y. et al. Hyaluronan Receptor LYVE-1-Expressing Macrophages Maintain Arterial Tone through Hyaluronan-Mediated Regulation of Smooth Muscle Cell Collagen. *Immunity* 49, 326–341.e7 (2018).
9. Soneson, C. & Robinson, M. D. Bias, robustness and scalability in single-cell differential expression analysis. *Nat. Methods* 15, 255–261 (2018).
10. Ferreira, P. G. et al. The effects of death and post-mortem cold ischemia on human tissue transcriptomes. *Nat. Commun.* 9, 490 (2018).
11. Madissoon, E. et al. scRNA-seq assessment of the human lung, spleen, and esophagus tissue stability after cold preservation. *Genome Biol.* 21, 1 (2019).

Reviewer Reports on the First Revision:

Referee #1 (Remarks to the Author):

The authors have been responsive to the addressable concerns I had raised. Below I list a few unresolved comments.

Please cite prior art:

Wang L et al, Single-cell reconstruction of the adult human heart during heart failure and recovery reveals the cellular landscape underlying cardiac function. [REDACTED] volume 22, pages 108–119(2020).

Tucker NR et al, Transcriptional and Cellular Diversity of the Human Heart. [REDACTED]. 2020 May 14. doi: 10.1161/[REDACTED].119.045401.

Among immune cells, blood contamination of tissue is a real potential for cofounders. The authors state this was addressed by comparing to resident cells from previous studies (response 6 to reviewer 2). First, this would need these prior data sets to be clean of contaminating cells. Second, it appears a bit odd to rely on data from studies that may have a similar problem. Third, your data indicate that currently contamination may be an issue. The healthy myocardium is expected to be contain few monocytes (based on prior FACS data) whereas you list monocytes as numerous. One other, orthogonal approach I have seen is to use blood data sets to remove contamination. This relies on the notion that many immune cells change phenotype once they enter tissues.

Page 2, Summary: "We define the complexity of the cardiac vascular network and identify cardiac resident macrophages with inflammatory and reparative transcriptional signatures." I recommend revising this sentence as the first part seems diffuse and the second claims novelty where there is none. These subsets have been previously identified and described in a number of publications. On this point, the authors mention in their rebuttal that it is not clear to them which prior work describes heart specific macrophages and their roles, functions and phenotypes. Below I provide a list of key publications which may serve as an introduction to this field. I am aware that they can't all be cited but they may help to put some of the immune cell data into better context.

Review from key expert:

Chen B, Brickshawana A, Frangogiannis NG. The Functional Heterogeneity of Resident Cardiac Macrophages in Myocardial Injury(CCR2+ Cells Promote Inflammation, Whereas CCR2- Cells Protect). [REDACTED]. 2019 Jan 18;124(2):183-185. doi: 10.1161/[REDACTED].118.314357. PubMed PMID: 30653429; PubMed Central PMCID: PMC6338423.

Recent scRNAseq paper describing heart mac subsets in mouse:

Chakarov S, Lim HY, Tan L, Lim SY, See P, Lum J, Zhang XM, Foo S, Nakamizo S, Duan K, Kong WT, Gentek R, Balachander A, Carbajo D, Bleriot C, Malleret B, Tam JKC, Baig S, Shabeer M, Toh SES, Schlitzer A, Larbi A, Marichal T, Malissen B, Chen J, Poidinger M, Kabashima K, Bajenoff M, Ng LG, Angeli V, Ginhoux F. Two distinct interstitial macrophage populations coexist across tissues in specific subtissular niches. [REDACTED]. 2019 Mar 15;363(6432). pii: eaau0964. doi: 10.1126/science.aau0964. PubMed PMID: 30872492.

Paper on mouse and human resident macs in heart conduction:

Hulsmans M, Clauss S, Xiao L, Aguirre AD, King KR, Hanley A, Hucker WJ, Wülfers EM, Seemann G, Courties G, Iwamoto Y, Sun Y, Savol AJ, Sager HB, Lavine KJ, Fishbein GA, Capen DE, Da Silva N, Miquerol L, Wakimoto H, Seidman CE, Seidman JG, Sadreyev RI, Naxerova K, Mitchell RN, Brown D, Libby P, Weissleder R, Swirski FK, Kohl P, Vinegoni C, Milan DJ, Ellinor PT, Nahrendorf M. Macrophages Facilitate Electrical Conduction in the Heart. [REDACTED]. 2017 Apr 20;169(3):510-522.e20. doi: 10.1016/[REDACTED].2017.03.050. PubMed PMID: 28431249; PubMed Central PMCID: PMC5474950.

Macrophage subset function:

Li W, Hsiao HM, Higashikubo R, Saunders BT, Bharat A, Goldstein DR, Krupnick AS, Gelman AE, Lavine KJ, Kreisel D. Heart-resident CCR2(+) macrophages promote neutrophil extravasation through TLR9/MyD88/CXCL5 signaling. [REDACTED]. 2016 Aug 4;1(12). pii: 87315. doi: 10.1172/[REDACTED].87315. PubMed PMID: 27536731;

Paper on the role of resident heart macs in development:

Leid J, Carrelha J, Boukarabila H, Epelman S, Jacobsen SE, Lavine KJ. Primitive Embryonic Macrophages are Required for Coronary Development and Maturation. [REDACTED]. 2016 May 13;118(10):1498-511. doi: 10.1161/[REDACTED].115.308270. Epub 2016 Mar 23. PubMed PMID: 27009605

Paper describing subsets in mouse, key markers CCR2 and MHCII. These are later confirmed in humans (your reference 62)

Epelman S, Lavine KJ, Beaudin AE, Sojka DK, Carrero JA, Calderon B, Brija T, Gautier EL, Ivanov S, Satpathy AT, Schilling JD, Schwendener R, Sergin I, Razani B, Forsberg EC, Yokoyama WM, Unanue ER, Colonna M, Randolph GJ, Mann DL. Embryonic and adult-derived resident cardiac macrophages are maintained through distinct mechanisms at steady state and during inflammation. [REDACTED]. 2014 Jan 16;40(1):91-104. doi: 10.1016/[REDACTED].2013.11.019. PubMed PMID: 24439267; PubMed. Central PMCID: PMC3923301.

First description of the abundance of heart resident macrophages, started the field: Pinto AR, Paolicelli R, Salimova E, Gospocic J, Slonimsky E, Bilbao-Cortes D, Godwin JW, Rosenthal NA. An abundant tissue macrophage population in the adult murine heart with a distinct alternatively-activated macrophage profile. [REDACTED]. 2012;7(5):e36814. doi: 10.1371/[REDACTED].0036814. Epub 2012 May 10.

Page 17, Neuronal populations

"Cardiac neuronal cells constitute the cardiac conduction system, composed of sinoatrial and atrioventricular nodes and Purkinje fibers, which propagate ventricular-activating wave-fronts, and sympathetic and parasympathetic ..." As far as I know, the cardiac conduction system does not contain neuronal cells. Rather, these cells are considered specialized conducting myocytes. Please consider revising. Perhaps this cluster can be renamed "cells of conduction system". It would be interesting to specifically compare them to neurons and report any parallels, if space permits.

Referee #2 (Remarks to the Author):

I have no further concerns.

Author Rebuttals on the First Revision

Point-by-point response to Reviewer 1

"The authors have been responsive to the addressable concerns I had raised. Below I list a few unresolved comments.

Please cite prior art:

Wang L et al, Single-cell reconstruction of the adult human heart during heart failure and recovery reveals the cellular landscape underlying cardiac function. Nature Cell Biology volume 22, pages 108–119(2020).

Tucker NR et al, Transcriptional and Cellular Diversity of the Human Heart. Circulation. 2020 May 14. doi: 10.1161/CIRCULATIONAHA.119.045401."

We thank the reviewer for this suggestion and have included these references now in the **Introduction**, page 3, as follows:

"Single-cell and single-nucleus transcriptomics (scRNA-Seq, snRNA-Seq) and multiplex smFISH imaging enable us to address anatomical specificities, molecular

signatures, intercellular networks and spatial relationships at unprecedented resolution². These technologies illuminate the coordinated communication of cardiac cells within their microenvironments that enable electromechanical connectivity, biophysical interactions, and signaling required for tissue homeostasis, which have been addressed in the past mostly in the context of model organisms, with limited studies of human samples^{3,4,5}.”

Among immune cells, blood contamination of tissue is a real potential for cofounders. The authors state this was addressed by comparing to resident cells from previous studies (response 6 to reviewer 2). First, this would need these prior data sets to be clean of contaminating cells. Second, it appears a bit odd to rely on data from studies that may have a similar problem. Third, your data indicate that currently contamination may be an issue. The healthy myocardium is expected to be contain few monocytes (based on prior FACS data) whereas you list monocytes as numerous.

We agree with the reviewer that it is important to take into consideration the possible blood origin of the immune cell populations. Previously, we based our interpretation of tissue-resident *versus* circulating populations on markers described in the literature.

We now add integration with published 10K PBMC scRNAseq data from 10X Genomics (https://support.10xgenomics.com/single-cell-gene-expression/datasets/3.0.0/pbmc_10k_v3) using the same logistic regression modeling (described in the **Methods** section of our manuscript) applied to the comparison of heart cell populations with kidney and skeletal muscle. This analysis yields the matches of cell populations shown in **Reviewer Table 1** below.

Reviewer Table 1. Projection of cardiac immune cell clusters to PBMC scRNA-Seq data by logistic regression.

The logistic regression models described in our cross-tissue analysis are used for projection onto PMBC scRNA-Seq data, highlighting the matches of our immune cells with lymphocyte and monocyte populations, and the absence of a blood match for the cardiac macrophage populations.

Immune Cell States of the Heart	Abundance in the Heart	Predicted abundance in the PBMC data [% of all PBMC]
CD16+Mo	3278	1791 (22.4 %)
B_cells	1195	1747 (21.8 %)
CD8+T_cytox	2956	660 (8.26 %)
NK	3628	491 (6.1 %)
Mo_classic	1883	475 (5.9 %)
CD4+T_cytox	3113	410 (5.1 %)
Mo_pi	1652	81 (1.1 %)
dpT	814	45 (0.5 %)
LYVE1+MΦ1	3018	36 (0.4 %)
NKT	1463	18 (0.2 %)
NΦ	121	0
CD4+T	1047	0
doublets	623	0
CD8+T_tem	3001	1 (0.01%)
IL17RA+Mo	32	0
LYVE1+MΦ2	2088	0
LYVE1+MΦ3	1957	0
Mast	1543	0
MΦ_AgP	1278	0
MΦ_mod	1313	0
DOCK4+MΦ1	3239	0
DOCK4+MΦ2	1626	0

This table shows that CD16+monocytes (CD16+Mo), classic monocytes (Mo_classic) and pro-inflammatory monocytes (Mo_pi) are very similar to circulating monocyte populations found in PBMCs, and suggest that they are likely circulatory. In contrast, the macrophage populations that we previously highlighted as tissue residents have 0 matches in the PBMC scRNA-Seq data, providing further support to the notion that they are *bona fide* tissue resident cells.

Please note that since *doublets* are a mixture of cells and they do not have a clear signature, it is expected that the model will not be able to provide a clear match. For CD4+ T cells, the model may not be able to differentiate between activated and memory phenotypes given the fluid transcriptional signature of these cells; so the absence of these cells is an artefact of the lack of sensitivity in the computational matching across CD4+ T cell subtypes. Also, it is known that circulating neutrophils (NΦ)

are not captured by most current droplet protocols, explaining the absence of this population in the blood.

We have added this data to the Supplementary Material as **Supplementary Table E5**, and have now stated in the text that based on this blood/heart data comparison, these monocytes are indeed most likely circulatory. This is in the *Immune cells and cardiac homeostasis* section of the maintext:

“While monocytes are abundant in our data; in agreement with the previous literature⁶⁶, these are likely from the circulation, supported by computational integration of our cardiac data with published PBMC scRNAseq data (**Supplementary Table E5**).”

One other, orthogonal approach I have seen is to use blood data sets to remove contamination. This relies on the notion that many immune cells change phenotype once they enter tissues.

We agree, and have described our approach to achieve this in point 2 above. We use logistic regression models to evaluate the similarity of our cardiac immune cells with those observed in blood from publicly available resources. These data confirm that monocytes are circulating as expected, and our macrophage populations are most likely resident.

“Page 2, Summary: “We define the complexity of the cardiac vascular network and identify cardiac resident macrophages with inflammatory and reparative transcriptional signatures.” I recommend revising this sentence as the first part seems diffuse and the second claims novelty where there is none. These subsets have been previously identified and described in a number of publications. On this point, the authors mention in their rebuttal that it is not clear to them which prior work describes heart specific macrophages and their roles, functions and phenotypes. Below I provide a list of key publications which may serve as an introduction to this field. I am aware that they can’t all be cited but they may help to put some of the immune cell data into better context.

Review from key expert:

Chen B, Brickshawana A, Frangogiannis NG. The Functional Heterogeneity of Resident Cardiac Macrophages in Myocardial Injury(CCR2+ Cells Promote Inflammation, Whereas CCR2- Cells Protect). Circ Res. 2019 Jan 18;124(2):183-185. doi: 10.1161/CIRCRESAHA.118.314357. PubMed PMID: 30653429; PubMed Central PMCID: PMC6338423.

Recent scRNAseq paper describing heart mac subsets in mouse:

Chakarov S, Lim HY, Tan L, Lim SY, See P, Lum J, Zhang XM, Foo S, Nakamizo S, Duan K, Kong WT, Gentek R, Balachander A, Carbajo D, Bleriot C, Malleret B, Tam JKC, Baig S, Shabeer M, Toh SES, Schlitzer A, Larbi A, Marichal T, Malissen B, Chen J, Poidinger M, Kabashima K, Bajenoff M, Ng LG, Angeli V, Ginhoux F. Two distinct interstitial macrophage populations coexist across tissues in specific subtissular niches. Science. 2019 Mar 15;363(6432). pii: eaau0964. doi: 10.1126/science.aau0964. PubMed PMID: 30872492.

Paper on mouse and human resident macs in heart conduction:

Hulsmans M, Clauss S, Xiao L, Aguirre AD, King KR, Hanley A, Hucker WJ, Wülfers EM, Seemann G, Courties G, Iwamoto Y, Sun Y, Savol AJ, Sager HB, Lavine KJ, Fishbein GA, Capen DE, Da Silva N, Miquerol L, Wakimoto H, Seidman CE, Seidman JG, Sadreyev RI, Naxerova K, Mitchell RN, Brown D, Libby P, Weissleder R, Swirski FK, Kohl P, Vinegoni C, Milan DJ, Ellinor PT, Nahrendorf M. *Macrophages Facilitate Electrical Conduction in the Heart*. *Cell*. 2017 Apr 20;169(3):510-522.e20. doi: 10.1016/j.cell.2017.03.050. PubMed PMID: 28431249; PubMed Central PMCID: PMC5474950.

Macrophage subset function:

Li W, Hsiao HM, Higashikubo R, Saunders BT, Bharat A, Goldstein DR, Krupnick AS, Gelman AE, Lavine KJ, Kreisel D. *Heart-resident CCR2(+) macrophages promote neutrophil extravasation through TLR9/MyD88/CXCL5 signaling*. *JCI Insight*. 2016 Aug 4;1(12). pii: 87315. doi: 10.1172/jci.insight.87315. PubMed PMID: 27536731;

Paper on the role of resident heart macs in development:

Leid J, Carrelha J, Boukarabila H, Epelman S, Jacobsen SE, Lavine KJ. *Primitive Embryonic Macrophages are Required for Coronary Development and Maturation*. *Circ Res*. 2016 May 13;118(10):1498-511. doi: 10.1161/CIRCRESAHA.115.308270. Epub 2016 Mar 23. PubMed PMID: 27009605

Paper describing subsets in mouse, key markers CCR2 and MHCII. These are later confirmed in humans (your reference 62):

Epelman S, Lavine KJ, Beaudin AE, Sojka DK, Carrero JA, Calderon B, Brija T, Gautier EL, Ivanov S, Satpathy AT, Schilling JD, Schwendener R, Sergin I, Razani B, Forsberg EC, Yokoyama WM, Unanue ER, Colonna M, Randolph GJ, Mann DL. *Embryonic and adult-derived resident cardiac macrophages are maintained through distinct mechanisms at steady state and during inflammation*. *Immunity*. 2014 Jan 16;40(1):91-104. doi: 10.1016/j.immuni.2013.11.019. PubMed PMID: 24439267; PubMed Central PMCID: PMC3923301.

First description of the abundance of heart resident macrophages, started the field:

Pinto AR, Paolicelli R, Salimova E, Gospocic J, Slonimsky E, Bilbao-Cortes D, Godwin JW, Rosenthal NA. *An abundant tissue macrophage population in the adult murine heart with a distinct alternatively-activated macrophage profile*. *PLoS One*. 2012;7(5):e36814. doi: 10.1371/journal.pone.0036814. Epub 2012 May 10."

We agree that there is a rich prior literature on cardiac macrophages, and further discussion of the literature would improve the manuscript despite space constraints. We have now incorporated some of these extra references into our **Discussion** section. We acknowledge that previous work has highlighted the presence of these myeloid phenotypes, mostly in mouse models, and that this is the first time we have described these populations so comprehensively in adult human hearts. For example, we identify the human counterpart of self-renewing, tissue-resident macrophages mediating cardiac remodelling after myocardial infarction described in mice (Dick et al. 2019). We also observe a population of macrophages expressing *LYVE1*, which has been associated with the regulation of collagen production in large vessels (Lim et al. 2018). Finally, we also identify a population of lipid-associated macrophages which mediate inflammation in a *TREM2*-dependent manner (Jaitin et al. 2019).

"We identify 21 different immune subpopulations, revealing immune cardiac myeloid and lymphoid cells and their interactions with FB and CM. Cardiac myeloid cells in our data overlap well with previous findings^{92–94}, provide new insights into the complexity of the macrophage repertoires from the six cardiac anatomical regions, allowing us to make inferences on paracrine circuits during cardiac homeostasis.

Through cross-tissue analysis with our human skeletal intercostal muscle data and previously published kidney data, we delineate cardiac populations distinct from those in skeletal muscle and kidney.”

For clarity we have also reworded the sentence in **Summary**, page 2:

“We define the complexity of the cardiac vasculature and its changes along the arterio-venous axis. In the immune compartment we identify cardiac-resident macrophages with inflammatory and protective transcriptional signatures.”

“Page 17, Neuronal populations

“Cardiac neuronal cells constitute the cardiac conduction system, composed of sinoatrial and atrioventricular nodes and Purkinje fibers, which propagate ventricular-activating wave-fronts, and sympathetic and parasympathetic ...” As far as I know, the cardiac conduction system does not contain neuronal cells. Rather, these cells are considered specialized conducting myocytes. Please consider revising. Perhaps this cluster can be renamed “cells of conduction system”. It would be interesting to specifically compare them to neurons and report any parallels, if space permits.”

We apologize for this poorly worded sentence. As both the cardiac conduction system and cells from the autonomic nervous system that regulate cardiac electrophysiology express neuronal transcripts, we use the term “neuronal cells” (NC) to describe both. For clarity, we have revised the header of this section as “**Cardiac Conduction System and Neuronal Cells**” and have rephrased the initial sentences on the page 17:

“The cardiac conduction system is composed of sinoatrial and atrioventricular nodes and Purkinje fibers, which propagate ventricular-activating wave-fronts, receive regulatory signals from sympathetic and parasympathetic components of the autonomic nervous system⁷⁵. Among the 3,961 cells that express prototypic electrophysiologic transcripts (NRXN1, NRXN3, KCNMB4)⁷⁶, we identify six NC subclusters...”

While cross-comparison with the brain cells would surely be interesting, many more conduction system cells would need to be isolated. As cardiac conduction system cells also reside in regions that were not sampled in the hearts studied here, we do not have a comprehensive data set on these populations, and so cannot address similarities and differences in this manuscript.

Editorial Note:

The final rebuttal from the authors was checked and approved by Referee #1